# Confidence-Aware Explanations for 3D Molecular Graphs via Energy-Based Masking

**Xufeng Liu** *xufeng.liu@stonybrook.edu*
*Stony Brook University*

**Wenhan Gao** *wenhan.gao@stonybrook.edu*
*Stony Brook University*

**Yi Liu** *yi.liu.4@stonybrook.edu*
*Stony Brook University*

**Reviewed on OpenReview:** *https://openreview.net/forum?id=V4oLqPOvJf*

## Abstract

Graph Neural Networks (GNNs) have become a powerful tool for modeling molecular data. To improve their reliability and interpretability, various graph explanation methods are proposed to identify key molecular substructures that drive model predictions. Many graph explainers introduce soft masks to enable gradient-based optimization, and then discretize the optimized masks to obtain explanatory subgraphs. While these methods perform well for 2D GNNs, there is a growing demand for 3D explanation techniques suited to 3D GNNs, which often surpass 2D GNNs in performance. However, existing explainers struggle with 3D GNNs because cutoff-based 3D graph construction yields denser graphs, with the number of edges growing quadratically with the number of atoms. Motivated by this, we identify key sources of explanation errors and derive an upper bound that decomposes the explanation error into two components: (i) the optimized soft-mask loss and (ii) the discrepancy introduced when discretizing the soft mask to form the explanatory subgraph. Our theoretical analysis shows that the second component is closely related to the soft-to-discrete mask gap and is amplified by graph density, making it particularly challenging for dense 3D graphs. To bridge this gap, we use an energy-based formulation and our method assigns two energy values to each atom, corresponding to importance and non-importance. The explanation model becomes more confident when the distinction between two states is clearer. By optimizing these energy values to distinguish the two cases, we minimize both components of the bound and identify a stable subgraph with high explanation fidelity. Experiments with various 3D backbone models on widely used datasets validate our method's effectiveness in providing accurate and reliable explanations for 3D molecular graphs. The code is publicly available at `https://github.com/xufliu/EDMA`.

## 1 Introduction

In recent years, molecular learning has emerged as a crucial area of study, driving advances in drug discovery, protein engineering, and materials science (Zhang et al., 2025; Fout et al., 2017; Liu et al., 2020b; Wu et al., 2018; Gupta et al., 2021; Yan et al., 2022; Lin et al., 2023; Liu et al., 2025a; Gao et al., 2026a). Traditionally, molecules have been represented as 2D planar graphs, where atoms serve as nodes and chemical bonds are depicted as edges without considering the geometric configurations. The limitations of 2D representations in capturing molecular properties have led to a growing focus on 3D graph representations (Schütt et al., 2017; Liu et al., 2022; Wang et al., 2022; Satorras et al., 2021; Gasteiger et al., 2021; Qu et al., 2026; Gao et al., 2026b; Wan et al., 2026) that represent entities with spatial coordinates, enabling them to capture complex spatial dependencies that are critical for tasks involving 3D molecular structures. This shift is

critical because the 3D structure of molecules, particularly their spatial arrangement, directly influences their chemical behaviors and biological functions. In response, 3D GNNs have been rapidly developed and shown to outperform their 2D counterparts in numerous tasks (Schütt et al., 2017; Anderson et al., 2019; Batzner et al., 2022; Satorras et al., 2021; Liu et al., 2022; Wang et al., 2022; Jamasb et al., 2024; Subedi et al., 2024).

As GNNs have shown great results in molecular learning, the need for explainability and interpretability has become increasingly important. Molecular systems are inherently complex, and GNNs are often treated as black-box models, making it difficult to understand how specific structural features contribute to predictions, which raises significant concerns regarding transparency in the decision-making process. GNN explanation methods aim to illuminate these decision-making processes by identifying key substructures of the graph (the molecule) that influence the model's predictions. Groundbreaking research has addressed these challenges, advancing our understanding of graph learning mechanisms across various contexts (Ying et al., 2019; Yuan et al., 2020; Luo et al., 2020; Pope et al., 2019; Huang et al., 2023; Qu et al., 2025; Liu et al., 2025b).

While existing explanation methods are effective for 2D GNNs, there is an urgent need for explanation techniques specifically designed for 3D GNNs. Current methods face challenges with 3D GNNs due to the construction of edges based on cut-off distances, leading to a quadratically large number of edges (Schütt et al., 2017; Gasteiger et al., 2020a; Liu et al., 2022; Wang et al., 2022; Satorras et al., 2021). Our experiments in Sec. 4 clearly demonstrate the inability of 2D methods to effectively explain 3D GNNs. In our study, we identify the sources of errors in explanations and break them down into two components, informed by a derived upper bound that relates the optimized masks to the actual subgraph. This gap is particularly significant for 3D GNNs due to the large volume of edges involved. In 2D GNN explanations, there are typically at most a few edges with mask values around 0.5, indicating uncertainty about their inclusion or exclusion in the final explanatory graph. However, in 3D GNN explanations, the number of uncertain edges grows rapidly, leading to suboptimal results and ambiguous explanation results that complicate the decision-making process rather than explaining it.

To enhance explanation fidelity, our method aims to bridge this gap by assigning two energy values to each atom in a 3D molecular graph. One energy reflects the scenario where this node is important in making the decision, while the other represents the scenario where it is unimportant. Drawing an analogy to physics, we assert that nodes with lower energy values correspond to greater stability in the explanatory results; thereby, we are more confident about the scenario with which it is associated. Current explanation models (Ying et al., 2019; Luo et al., 2020; Miao et al., 2022) only optimize the first term in our derived bound leading to *over-smoothing of the soft masks*. Our approach seeks to push the lower energy down and push the higher energy up to simultaneously optimize both components of the derived error bound, thereby reducing discrepancies between the identified explanatory subgraph and the associated edge masks. By achieving a lower energy state, we can accurately and confidently identify a stable subgraph that exhibits high explanation fidelity. An illustration of our method to mitigate challenges caused by the key structural differences between 2D and 3D GNN explanations is presented in Fig. 1. Experiments are conducted on several backbone networks and two widely used datasets, the QM9 (Ramakrishnan et al., 2014) and the EC (Wang et al., 2023), validate the efficacy of our method, demonstrating its capacity to deliver accurate, stable, and reliable explanations for 3D molecular graphs.

We summarize our contributions as follows:

- Leveraging the structural differences between 2D and 3D GNNs, we reformulate graph explanations specifically for 3D GNNs.
- We establish a theoretical error bound for 3D graph explanations and decompose the explanation error into two components: (i) the optimized soft mask loss and (ii) the soft-to-discrete discrepancy that has been largely overlooked.
- Based on this bound, we introduce Energy-based Discrete Mask Approximation (EDMA) to reduce both components simultaneously, and provide theoretical analysis showing that EDMA tightens the discrepancy term in dense 3D graphs.
- Experimental results demonstrate that our method is effective and highly generalizable for explaining 3D GNNs.

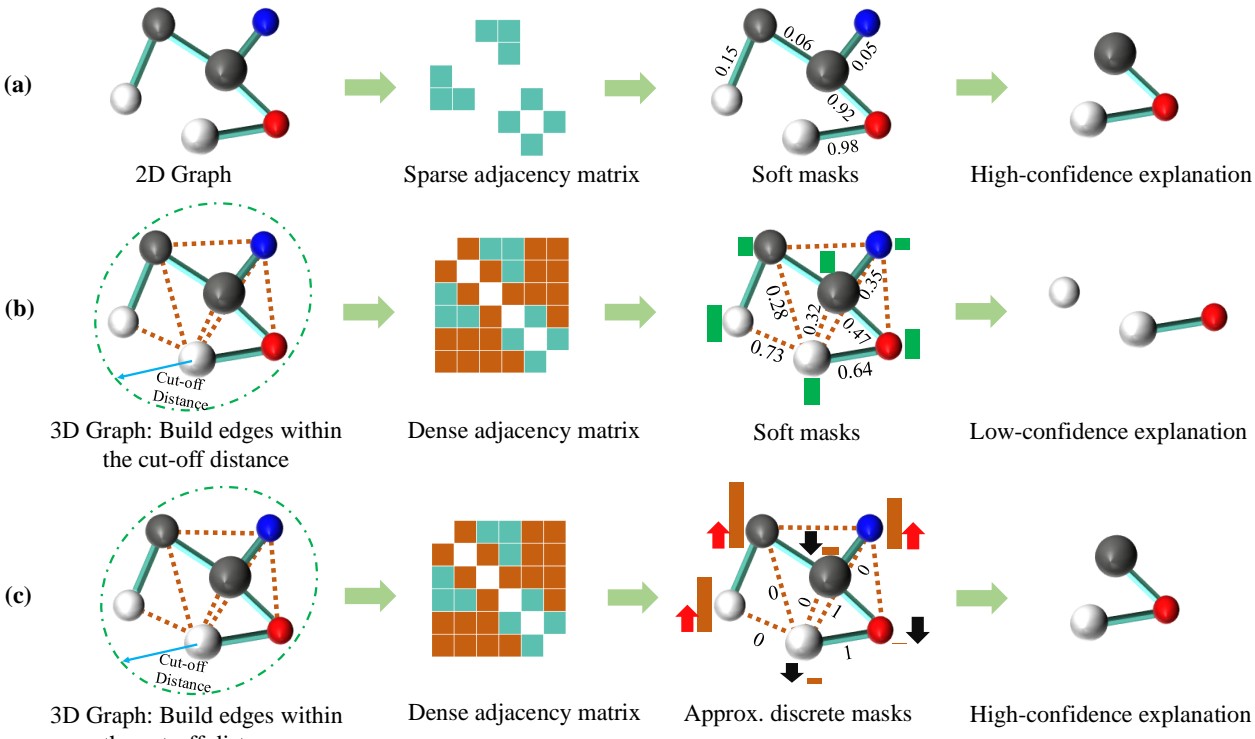

Figure 1: An illustration of the structural differences between 2D and 3D GNN explanations, as well as the challenges posed by existing methods. In the third column, the bars represent the energies for nodes and the arrows represent the operations to "push up" and "push down" the energies, while the numbers correspond to the edge masks derived from the energies of the nodes. Specifically, since 3D GNN edges are induced by geometry, we explain predictions by selecting a subset of nodes. However, using soft masks typically results in explanations of low confidence, which undermines explanation fidelity. To address this issue, we assign energies to each node. By pushing up and down these energies, we can obtain approximately discrete masks that provide more confident explanations. Further details are provided in Sec. 3.

## 2 Background and Related Work

In this section, we begin by presenting the formal definition of the graph explanation task in Sec. 2.1. Following that, in Sec. 2.2, we provide a comprehensive review of the key methodologies that have been proposed to generate explanations. Finally, Sec. 2.3 delves into the definition and formulation of Energy-Based Models (LeCun et al., 2006), outlining their role in enhancing the interpretability of GNNs and their applications in providing insights into molecular structures and behaviors.

### 2.1 Graph Explanation

A 2D molecular graph $G$ is represented as $G = (\mathcal{V}, \boldsymbol{X}, \mathbf{A})$, where $\mathcal{V} = \{v_1, v_2, \ldots, v_n\}$ denotes a set of $n$ nodes. $\boldsymbol{X} = [\mathbf{x_1}, \mathbf{x_2}, \ldots, \mathbf{x_n}]^T \in \mathbb{R}^{n \times d_v}$ is the node feature matrix with each $\mathbf{x_i} \in \mathbb{R}^{d_v}$, where $d_v$ represents the dimension of the node features. $\mathbf{A} = \{a_{ij} \mid i, j \in \mathcal{V} \text{ and } i \neq j\} \in \{0, 1\}^{n \times n}$ is the adjacency matrix, where each $a_{ij} \in \{0, 1\}$ denotes whether there is an edge from node $i$ to node $j$. Graph neural networks (GNNs) utilize the edge set $\mathbf{A}$ to facilitate message passing and aggregation between nodes. A GNN $\Phi$ is a mapping from a graph $G$ to a prediction $\hat{Y}$ in relation to the target variable $Y$. This target can represent discrete labels in a graph classification task or continuous values in a regression task. In this study, we present the method in the graph regression setting without loss of generality. The experiments in Sec. 4 cover both regression and classification tasks.

**Graph Explanation:** Following the definition in Ying et al. (2019), the objective of instance-level graph explanation is to identify a subgraph $G_S \subseteq G$ that is important to the target $Y$. This is formally expressed

as:

$$G_S^* = \underset{G_S \subseteq G}{\arg\min} \, \mathcal{L}(Y; \Phi(G_S)) \quad \text{s.t.} \quad |G_S| \leq B, \tag{1}$$

where $\mathcal{L}$ denotes the task-dependent loss function, and $B$ represents a size constraint on the subgraph to avoid trivial solutions. Eq. (1) can be rewritten as:

$$G_S^* = \underset{\mathbf{M}}{\arg\min} \, \mathcal{L}(Y; \Phi(\mathbf{X}, \mathbf{M} \odot \mathbf{A}))$$
$$\text{s.t.} \quad \mathbf{M} \in \{0,1\}^{n \times n}, \quad \sum_{i=1}^{n} \sum_{j=1}^{n} \mathbf{M}_{ij} \leq B. \tag{2}$$

Here, $\odot$ denotes the element-wise (Hadamard) product. Directly solving Eq. (2) leads to a computationally intractable combinatorial optimization problem with complexity $O(2^n)$. Existing works relaxed the discrete (hard) masks with discrete values 0 and 1 to soft masks with values between 0 and 1:

$$G_S^* = \underset{\mathbf{M}'}{\arg\min} \, \mathcal{L}(Y; \Phi(\mathbf{X}, \mathbf{M}' \odot \mathbf{A}))$$
$$\text{s.t.} \quad \mathbf{M}' \in [0,1]^{n \times n}, \quad \sum_{i=1}^{n} \sum_{j=1}^{n} \mathbf{M}'_{ij} \leq B. \tag{3}$$

Such relaxation enables gradients to be back-propagated; thus, gradient descent can be used to efficiently solve this problem.

## 2.2 Existing Explanation Methods

GNN explanation methods can be classified according to several criteria: transductive or inductive explanations, instance-level explanations (Ying et al., 2019; Schlichtkrull et al., 2021) versus model-level explanations (Yuan et al., 2020), model-specific approaches (Pope et al., 2019; Dai & Wang, 2021) compared to model-agnostic methods (Luo et al., 2020), and node-level (Ying et al., 2019; Pope et al., 2019) versus subgraph-level explanations (Yuan et al., 2021; Qu et al., 2025). In terms of explanation strategies, four primary categories emerge: (1) Gradient-based methods (Zhou et al., 2016) compute the gradients of target predictions with respect to inputs via back-propagation but often impose structural constraints on GNNs; (2) Decomposition methods (Pope et al., 2019) assign importance scores to input features by analyzing model parameters to reveal relationships between inputs and outputs; (3) Surrogate methods (Huang et al., 2023) use interpretable models to explain the behavior of complex GNNs, though they often encounter difficulties with the discrete and topological nature of 3D graphs; (4) Perturbation-based methods (Ying et al., 2019; Luo et al., 2020) identify important subgraphs by perturbing edges or nodes with masks and analyzing output prediction changes. Many perturbation-based explainers rely on the soft-mask optimization in Eq. (3) with a budget constraint and (often) an entropy regularizer to encourage compact and near-binary masks; see Sec. 3.4.

## 2.3 Energy-based Model

The central concept of Energy-Based Models (EBMs) is the use of Boltzmann distributions to assess the likelihood of input samples. This involves defining a function $\mathcal{E}(\mathbf{x_i}) : \mathbb{R}^{d_v} \to \mathbb{R}$ that assigns a non-probabilistic scalar known as *energy* to each configuration of the input data. Influential works (Xie et al., 2016) have significantly shaped research in this domain. EBMs have achieved notable success in various applications, including classification (Li et al., 2022; Grathwohl et al., 2019), regression tasks (Danelljan et al., 2020), structured prediction (Rooshenas et al., 2019; Belanger & McCallum, 2016), and out-of-distribution (OOD) detection (Liu et al., 2020a). Additionally, (Pang & Wu, 2021) have explored the use of EBMs in latent space for generation, while others have applied them to unsupervised learning (Ranzato et al., 2007) and concept-based modeling (Xu et al., 2024).

# 3 Energy-based Discrete Mask Approximation

In this section, we present the Energy-based Discrete Mask Approximation (EDMA), a principled approach for 3D GNN explanation. We begin by analyzing the differences between 2D and 3D graph explanations in Sec. 3.1. With such differences, we reformulate 3D graph explanations and identify an upper bound on the

explanation loss in Sec. 3.2. In Sec. 3.3, we provide a detailed presentation of our method to simultaneously optimize all terms in the derived upper bound. The key differences between EDMA and regularization-based approaches are discussed in Sec. 3.4.

## 3.1 2D v.s. 3D Graph Explanation

The primary distinction between 2D and 3D graph explanations arises from the structural differences inherent in 2D and 3D graphs. In 2D graphs, each node is associated with a set of edges that connect these nodes in a planar layout. In 3D graphs, nodes are represented in three-dimensional space, allowing for a more accurate depiction of the physical arrangement and spatial relationships between entities, while edges are typically determined from the coordinates of the nodes by a cut-off distance (Schütt et al., 2017; Gasteiger et al., 2020a; Liu et al., 2022; Wang et al., 2022).

More concretely, for small molecular structures, the number of bonds (edges) between atoms (nodes) is typically limited, resulting in rather sparse graphs. However, 3D GNNs do not utilize chemical bonds as edges; instead, the 3D spatial configurations of nodes are used to construct edges, resulting in a quadratically large number of edges. As a result, 3D GNN explanation poses significant challenges to existing methods, partially illustrated in Fig. 1 and detailed below:

1. **Edge coupling in 3D cutoff graphs**: Many perturbation-based explanation methods optimize edge masks by treating each edge as an independent decision variable, which is a common design choice in widely used 2D explainers such as GNNExplainer and PGExplainer (Ying et al., 2019; Luo et al., 2020). This design is often reasonable for sparse 2D graphs, where edges correspond to explicit chemical bonds. However, in 3D molecular graphs, *edges are typically induced from geometry* (e.g., cutoff neighbors) and mainly reflect spatial proximity and overlapping local environments. Consequently, edges are strongly coupled: selecting or removing one atom simultaneously affects many edges, and neighboring edges share overlapping geometric neighborhoods rather than independent "chemical bonds".

2. **Dense Adjacency Matrix**: The adjacency matrix $\mathbf{A}$ in a 3D graph is typically dense, unlike the sparse adjacency matrix commonly observed in a 2D graph. As one can imagine, this leads to a problem of combinatorial complexity with respect to the number of edges with discrete masks. Even with soft mask relaxation, the large number of edges introduces a substantial lack of confidence in identifying the explanatory subgraph, often resulting in suboptimal explanation outcomes. Specifically, with low confidence in distinguishing important and unimportant sub-parts, the soft-masked "subgraph" deviates a lot from the final discrete explanatory subgraph. The optimization process might find a soft-masked "subgraph" with minimal loss in Eq. (3); however, when we decide the explanatory subgraph from soft-masks, the lack of confidence in the soft masks leads to poor final explanation performance.

## 3.2 Reformulating 3D Graph Explanation

The inputs to 3D GNNs consist of nodes with 3D spatial coordinates, and edges are constructed based on this spatial information. The common assumption of edge independence no longer holds in this context. Simply applying current graph explanation methods without accounting for the structural differences is unlikely to yield explanations of scientific meanings, as demonstrated in our experiments in Sec. 4 where methods not specifically designed for 3D GNNs yield worse performance.

*As discussed in the first challenge outlined in Sec. 3.1, we should define the explanatory substructure to be a subset of nodes.* To this end, we place masks on the nodes, which are then transformed into edge masks. For node $i$, there will be an associated soft-mask value $m_i \in [0, 1]$, and $\mathbf{m} \in [0, 1]^n$ denotes the set of all node masks. The edge masks are then constructed by $\mathbf{M}' = \mathbf{m} \otimes \mathbf{m}$, where $\otimes$ denotes the outer product. Specifically, Eq. (3) can be rewritten as:

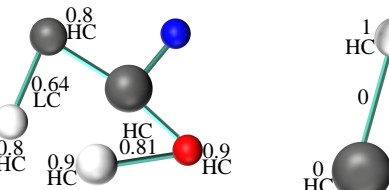

**Soft mask:** Low Confidence    **Discrete mask:** High Confidence

Figure 2: A comparison between soft masks and discrete masks. Numbers are mask values, **HC** denotes High Confidence, and **LC** denotes Low Confidence. The edge masks used for message passing are constructed from node masks. However, soft masks can lead to great discrepancies between the optimization objective and the final explanatory subgraph, as indicated in Eq. (5).

$$G_S^* = \underset{\mathbf{M}'}{\arg\min} \mathcal{L}(Y; \Phi(\mathbf{X}, \mathbf{M}' \odot \mathbf{A}))$$

$$\text{s.t.} \quad \mathbf{m} \in [0,1]^n, \quad \mathbf{M}' = \mathbf{m} \otimes \mathbf{m}, \quad \|\mathbf{m}\|_1 \leq K, \tag{4}$$

where $\|\mathbf{m}\|_1 = \sum_{i=1}^n \mathbf{m}_i$ is the 1-norm of $\mathbf{m}$, $K$ is the budget on the number of nodes in the final explanatory substructure. To this end, we would like to introduce a shortcoming in relaxing discrete masks to soft-masks: The discrepancy between the optimized soft-masked "subgraph" and the final explanatory subgraph. Mathematically,

$$G_S^* = \underset{G_S \subseteq G}{\arg\min} \mathcal{L}\big(Y; \Phi(G_S)\big) \leq \underbrace{\mathcal{L}\big(Y; \Phi(\mathbf{X}, \mathbf{M}' \odot \mathbf{A})\big)}_{\text{soft-mask explanation loss}}$$

$$+ \underbrace{\mathcal{L}\big(\Phi(\mathbf{X}, \mathbf{M}' \odot \mathbf{A}); \Phi(\mathbf{X}, \mathbf{M} \odot \mathbf{A})\big)}_{\text{discrepancy between soft and discrete masks}}. \tag{5}$$

In this bound, the first term represents the soft-mask explanation loss in the relaxed optimization Eq. (3), which we solve through gradient descent. The second term depicts the discrepancy between soft and discrete masks, and this has been overlooked in existing GNN explanation methods. Existing studies term this issue as "introduced evidence". Any value in masks that is not strictly zero or one can introduce new semantics or noise into the explanation, potentially impacting the results (Dabkowski & Gal, 2017). For instance, even if the value of $\mathbf{M}'_{ij}$ is small, the edge $e_{ij}$ may still facilitate message passing between node $i$ and $j$. We will refer to this as the confidence of the soft masks, where a mask value close to 0 or 1 indicates high confidence about the substructure's contribution to the decision-making process. Appendix A formalizes this phenomenon by showing that the soft-to-discrete gap scales with (i) the soft-to-hard distance of the node mask and (ii) graph density (node degrees). This implies that simply optimizing soft masks can be intrinsically unreliable in dense 3D graphs unless the method explicitly enforces discreteness—precisely what our method is designed to do. *While this bound generally applies to both 2D and 3D GNNs, 3D GNNs suffer much more from this issue for reasons given in the second challenge outlined in Sec. 3.1.* In 3D GNNs, there are quadratically many edges, and the accumulation of information passed during message passing can significantly influence the explanation results even with edge masks of high confidence. Even worse, due to the intrinsic nature of 3D GNNs, the node masks will largely decrease the confidence and stability in final explanations as illustrated in Fig. 2. To this end, we are ready to present the Energy-based Discrete Mask Approximation (EDMA) method to mitigate this issue.

### 3.3 EDMA for Confident 3D Graph Explanation

We now present our method EDMA for confident 3D graph explanation that simultaneously minimizes both terms in Eq. (5).

Instead of using soft masks for the selection of explanatory nodes, we treat the selection of nodes as states within a system, where the energy levels of these states determine their probability of being part of the explanatory subgraph. The EBM function $\mathcal{E}(\mathbf{e_i}) : \mathbb{R}^d \to \mathbb{R}$ maps the node embedding $\mathbf{e_i}$ to a scalar value known as *energy*. Following Liu et al. (2020a), the energy for a node with respect to class $c$ is defined as $\mathcal{E}(\mathbf{e_i}, c) = \mathcal{E}_c(\mathbf{e_i}) = \frac{-\phi_c(\mathbf{e_i})}{T}$, where $\phi_c$ extracts logits for class $c$ and $0 < T < 1$ serves as a control hyper-parameter analogous to the temperature in physics. It will push up the larger energy and push down the smaller energy. To see this, suppose $\phi_0 = 2$ and $\phi_1 = 5$ then the difference between them is 3. With $T = 0.1$, the energies will be 20 and 50, respectively, and the difference between them is 30. With a smaller value of $T$, we further amplify the difference

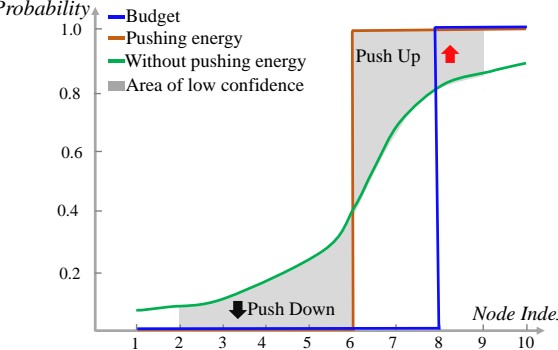

Figure 3: An illustration of the effects of the explainer function $f$. The node indices are arranged based on their probability values. By pushing up and down the energies, the masks become approximately discrete, enhancing confidence in the explanatory substructure. Moreover, varying the values of the stretching parameters ($\gamma, \zeta$ in Eq. (6)) enables us to better control the budget.

between these two energies, leading to a more confident explanation with our explainer as defined in Eq. (6). This process is analogous to the temperature in physics that, when $T$ is small, the system is in a low-energy state, leading to probabilities closer to 0 or 1; in other words, a more confident selection of nodes.

Then, an explainer function $f$ is used to compute the probabilities that each node $i$ belongs to the explanatory subgraph, represented as $P_i(c = 1) = f\big(\mathcal{E}_0(\mathbf{e_i}), \mathcal{E}_1(\mathbf{e_i})\big)$. Inspired by the hard concrete distribution (Louizos et al., 2017), $f : \mathbb{E} \times \mathbb{E} \mapsto [0, 1]$, with $\mathbb{E}$ being the potential energy state space for all nodes, is a function that takes both energies and produces a single scalar value indicating probabilities defined as:

$$
\begin{aligned}
&f\big(\mathcal{E}_0(\mathbf{e_i}), \mathcal{E}_1(\mathbf{e_i})\big) \\
&= \min\left(1, \max\left(0, \frac{1}{1 + e^{\mathcal{E}_0(\mathbf{e_i}) - \mathcal{E}_1(\mathbf{e_i})}}(\zeta - \gamma) + \gamma\right)\right),
\end{aligned}
\tag{6}
$$

where $\gamma < 0, \zeta > 1$ are hyper-parameters to stretch the probability to the interval $(\gamma, \zeta)$ and then truncate the value to the range $[0, 1]$. Together with the temperature, this will help us obtain more confident explanations in terms of having all probabilities close to 0 or 1 and better control the budget $K$ as we can set appropriate values of $\gamma$ and $\zeta$ to obtain the desired number of probabilities closer to 1. In Fig. 3, we illustrate how our energy-based explainer function generates approximately discrete masks while effectively managing the budget. Our explainer function takes energy values as input and outputs the probability that a node belongs to the explanatory substructure. By increasing the energies of important nodes and decreasing those of unimportant ones, we enhance their distinction, resulting in approximately discrete masks that yield more confident explanatory substructures. Furthermore, the stretching parameters in Eq. (6) regulate the number of nodes for which energies are pushed up.

Without loss of generality, we denote $m_i = f\big(\mathcal{E}_0(\mathbf{e_i}), \mathcal{E}_1(\mathbf{e_i})\big)$ as the explanation mask. Finally, the loss term in Eq. (4) and the explainer function in Eq. (6) are then jointly optimized to classify nodes and determine whether they belong to the explanatory subgraph. The final optimization function for our proposed method is as follows:

$$
\mathcal{L}_{final} = \mathcal{L}\big(Y; \Phi(\mathbf{X}, \mathbf{M}^{'} \odot \mathbf{A})\big) + \alpha \left\| f\big(\mathcal{E}_0(\mathbf{e_i}), \mathcal{E}_1(\mathbf{e_i})\big) \right\|_1,
\tag{7}
$$

where $\alpha$ is a parameter that balances the information loss and the explainer function loss that controls the budget. With this particular formulation, we simultaneously optimize both terms in our derived bound in Eq. (5), leading to approximately discrete, i.e., confident, probabilities for the inclusion or exclusion of a certain node in the final explanatory subgraph. Appendix B provides theoretical support for this design choice. In particular, it shows that EDMA's energy-gap gating admits a variational characterization as the unique minimizer of an entropy-regularized linear objective, and that the resulting soft-to-discrete discrepancy bound tightens monotonically as $T$ is annealed. This motivates enforcing *discreteness by design*, which EDMA achieves via margin-controlled energy gating and temperature annealing.

### 3.4 Discreteness by Design v.s. Regularization

A common alternative to promote discreteness is the introduction of regularization terms to constrain deviations of the masks from discreteness. This approach is used in most GNN explanation methods, e.g., GNNExplainer (Ying et al., 2019) and PGExplainer (Luo et al., 2020). Specifically, the optimization objective is:

$$
G_S^* = \arg\min_{\mathbf{M}^{'}} \mathcal{L}\big(Y; \Phi(\mathbf{M}^{'} \odot \mathbf{A}, \mathbf{X})\big) + \alpha \cdot \|\mathbf{M}^{'}\|_1 + \beta \cdot \mathbb{H}[\mathbf{M}^{'}],
\tag{8}
$$

where $\alpha$ and $\beta$ are coefficients that balance different losses. $\alpha \cdot \|\mathbf{M}^{'}\|_1$ enforces the budget and $\beta \cdot \mathbb{H}[\mathbf{M}^{'}]$ represents the entropy of the mask values, ensuring that they remain close to binary. Most existing explanation methods adopt this setting and include the entropy loss to promote discreteness.

However, this regularization-based approach has several limitations. First, the introduction of additional loss terms creates a trade-off between enforcing discreteness and optimizing the primary objective, requiring careful hyperparameter tuning to achieve an appropriate balance. Poorly chosen regularization strengths can lead to suboptimal solutions where the mask is neither sufficiently discrete nor effective in explaining the model's predictions. Second, regularization introduces additional complexity into the optimization process;

for example, competing loss terms can create conflicting gradients, making convergence slower and less stable. In contrast, our method enforces discreteness by design; it directly enforces binary mask constraints without relying on auxiliary regularization terms. These complications become worse for 3D graph explanations due to quadratically many edges, making the regularization-based approach highly ineffective for 3D graphs. As demonstrated in the ablation study in Sec. 4.4, removing the entropy loss does not result in noticeable performance degradation; the performance gains from our method significantly outweigh the degradation from the absence of entropy loss in existing explainers, demonstrating the superiority of our method as a principled approach to enforce discreteness by design.

**Remark.** Regularization-based approaches (e.g., $\ell_1/\ell_0$-style penalties or entropy penalties) primarily encourage *sparsity*, but do not directly control the *soft-to-hard distance* of the learned mask, which is the quantity amplified by dense 3D graphs. Stochastic relaxations such as hard-concrete or Gumbel-Softmax provide differentiable surrogates via random gates and are typically analyzed in expectation, which can still permit intermediate-probability masks during optimization. In contrast, EDMA enforces *discreteness by design* through an energy-gap parameterization and temperature annealing, directly targeting the soft-to-discrete discrepancy term in Eq. (5). A detailed discussion is provided in Appendix B.

## 4 Experimental Studies

We begin by outlining the experimental setup in Sec. 4.1. Sec. 4.2 presents a comparative analysis of the quantitative results of our method against baseline approaches. In Sec. 4.3, we offer a qualitative analysis to further illustrate the interpretability and effectiveness of the proposed method. Finally, Sec. 4.4 provides additional analyses on the contributions of different design choices.

### 4.1 Experimental Setup

**Dataset and Backbone 3D GNNs.** In this work, we conduct experiments on two widely used datasets—the QM9 (Ramakrishnan et al., 2014) dataset and the EC dataset (Wang et al., 2023). QM9 is a comprehensive 3D molecular dataset frequently used to predict various molecular properties; we use the version available in PyTorch Geometric (PyG) directly, along with its predefined training and test splits. Our study targets the prediction of two key properties: the dipole moment ($\mu$) and the free energy at 298.15K (denoted as $G_f$ to avoid confusion with the graph notation $G$). EC is a multi-classification task (60 classes) for *enzymes* that classifies enzyme commission (EC) numbers based on the reactions catalyzed by enzymes. Compared to the QM9 dataset, the EC dataset is a classification task, instead of regression, with larger and more complex graph structures, *demonstrating the generalization ability of our proposed method*. The EC dataset contains $5,481$ graphs, which are split into $4,149$ for training and $1,332$ for testing. We adopt the pretrained SchNet (Schütt et al., 2017) and DimeNet++ (Gasteiger et al., 2020b;a) from PyG as the backbone 3D GNN models for the QM9 dataset. The EC dataset is a multi-class classification task for *proteins*, whereas SchNet and DimeNet++ for the QM9 dataset are designed specifically for small molecules. Therefore, we adopt ProNet (Wang et al., 2023), a well-known 3D GNN for protein learning, as the backbone model.

**Baselines.** We compare our approach against several state-of-the-art baselines. GNNExplainer (Ying et al., 2019) and PGExplainer (Luo et al., 2020) are **leading explanation methods for 2D GNNs**, designed for transductive and inductive tasks, respectively. However, due to structural differences discussed in Sec. 3.1, these methods are not directly applicable to 3D GNNs. To adapt them for 3D molecular graphs, we place masks on nodes, generate edges in a manner similar to our approach, and use these to perturb node embeddings and generate explanations. These adapted methods, referred to as GNNExplainer-Dense and PGExplainer-Dense, serve as key baselines for evaluating our proposed method.

In addition, we include LRI (Miao et al., 2022) in our comparisons, as it is currently the only method specifically designed for geometric graph explanations. We employ the LRI-Bernoulli variant, which identifies key nodes relevant to downstream regression tasks, making it a strong baseline for explaining 3D molecular graph data. Details of the baseline methods are provided in Appendix C. All baseline methods are implemented using PyG with necessary adjustments to ensure consistency in the experimental setup. EDMA incurs negligible additional overhead and has the same asymptotic complexity as existing methods (Appendix D).

Table 1: Explanation Fidelity$^-$ (the lower the better) for all baseline methods and our proposed EDMA method using SchNet as the backbone on the property $\mu$ (dipole moment, in Debye (D)). The best results are highlighted in bold.

| Top-$k$ | 2 | 3 | 4 | 5 | 6 | 7 | 8 | 9 |
|---|---|---|---|---|---|---|---|---|
| GNNExplainer-Dense | 3.88 | 5.62 | 7.28 | 8.05 | 8.27 | 8.00 | 7.59 | 6.87 |
| PGExplainer-Dense | 2.91 | **3.73** | 4.83 | 6.09 | 6.62 | 6.55 | 6.81 | 6.08 |
| LRI-Bernoulli | 3.50 | 4.84 | 6.16 | 6.88 | 7.10 | 7.29 | 7.43 | 7.32 |
| **EDMA** | **2.74** | **3.73** | **4.31** | **4.83** | **5.08** | **5.47** | **5.72** | **5.31** |

Table 2: Explanation Fidelity$^-$ (the lower the better) for all baseline methods and our proposed EDMA method using SchNet as the backbone on the property $G_f$ (free energy, in eV). The results are presented in units scaled by a factor of 1000. The best results are highlighted in bold.

| Top-$k$ | 2 | 3 | 4 | 5 | 6 | 7 | 8 | 9 |
|---|---|---|---|---|---|---|---|---|
| GNNExplainer-Dense | 9.66 | 8.48 | 7.24 | 6.03 | 4.78 | 3.51 | 2.26 | 1.09 |
| PGExplainer-Dense | 10.26 | 9.56 | 8.68 | 7.74 | 6.52 | 5.40 | 4.21 | 3.94 |
| LRI-Bernoulli | 9.39 | 8.39 | 7.38 | 6.59 | 5.93 | 5.31 | 4.78 | 4.09 |
| **EDMA** | **8.66** | **7.45** | **6.23** | **5.07** | **3.74** | **2.55** | **1.36** | **0.21** |

**Evaluation Metrics.** Following standard practices, for the regression task with the QM9 data, we use Mean Absolute Error (MAE) to evaluate the performance of 3D GNN predictions against ground-truth molecular properties. A lower MAE indicates better results. In the context of explaining 3D molecular graphs, let $\text{MAE}_W$ represent the prediction error using the entire graph, while $\text{MAE}_S$ denotes the prediction error using the optimal subgraph selected for explanation, as described in Eq. (2). Naturally, $\text{MAE}_S$ is expected to be higher than $\text{MAE}_W$, as the complete graph generally yields better predictions than the subgraph, given the same pretrained 3D GNN model (such as SchNet or DimeNet++). We define explanation fidelity as $\text{Fidelity}^- = \text{MAE}_S - \text{MAE}_W$, which measures the quality of explanations produced by different methods. *A lower Fidelity$^-$ indicates that the method provides a more accurate and reliable explanation ($-$ indicates that lower is better).*

Similarly, for the classification task with the EC data, we use accuracy to evaluate the performance of 3D GNN predictions against ground-truth molecular labels. Following Faber et al. (2021), to ensure a more reliable comparison of results, we only select molecular graphs that have been correctly classified by the backbone model for explanation, thereby excluding the influence of misclassified instances. We define explanation fidelity Fidelity$^+$ as the percentage of explanatory subgraphs that produce the same labels as the original whole graphs. *A higher Fidelity$^+$ indicates that the method provides a more accurate and reliable explanation ($+$ indicates the higher the better).* Note that the reported results are averaged across all molecules in the test set. Since the standard deviation is two orders of magnitude smaller than the average fidelity, we omit it from our reported results. Further details about experimental settings are presented in Appendix E.

## 4.2 Quantitative Results

**Regression Task on the QM9 Dataset.** We conduct experiments to demonstrate the effectiveness of our method by comparing it to baseline approaches on the QM9 dataset. The results using SchNet are presented in Tables 1 and 2. Notably, all baseline methods select the top-$k$ nodes as explanations, with $k$ ranging from 2 to 9. Our method consistently outperforms these baselines, producing explanatory subgraphs with the lowest explanation fidelity (where lower fidelity indicates better performance). This suggests that by optimizing the two components based on the derived upper bound, we achieve a closer alignment between the desired discrete masks and the approximate discrete masks generated via the EBM. By appropriately controlling the loss function, we can either increase or decrease the energy of each atom, amplifying the distances between explanatory and non-explanatory parts, making them easier to identify. Similar results are observed in Tables 3 and 4 using DimeNet++ as the backbone. We note that the trend in Table 1 differs

Table 3: Explanation Fidelity$^-$ (the lower the better) for all baseline methods and our proposed EDMA method using DimeNet++ as the backbone on the property $\mu$ (dipole moment, in Debye (D)). The best results are highlighted in bold.

| Top-$k$ | 2 | 3 | 4 | 5 | 6 | 7 | 8 | 9 |
|---|---|---|---|---|---|---|---|---|
| GNNExplainer-Dense | 2.50 | 2.29 | 2.07 | 1.80 | 1.53 | 1.27 | 1.04 | 0.82 |
| PGExplainer-Dense | 2.52 | 2.46 | 2.13 | 2.07 | 1.84 | 1.67 | 1.50 | 1.40 |
| LRI-Bernoulli | 2.58 | 2.42 | 2.22 | 2.04 | 1.87 | 1.69 | 1.51 | 1.40 |
| **EDMA** | **2.40** | **2.07** | **1.76** | **1.47** | **1.22** | **1.02** | **0.85** | **0.71** |

Table 4: Explanation Fidelity$^-$ (the lower the better) for all baseline methods and our proposed EDMA method using DimeNet++ as the backbone on the property $G_f$ (atomization free energy, in eV). The best results are highlighted in bold.

| Top-$k$ | 2 | 3 | 4 | 5 | 6 | 7 | 8 | 9 |
|---|---|---|---|---|---|---|---|---|
| GNNExplainer-Dense | 65.74 | 62.56 | 58.91 | 54.65 | 49.15 | 43.18 | 36.69 | 30.53 |
| PGExplainer-Dense | 64.32 | 61.25 | 55.33 | 51.64 | 46.13 | 40.33 | 34.35 | 29.41 |
| LRI-Bernoulli | 64.76 | 60.46 | 55.29 | 49.97 | 44.66 | 39.41 | 34.19 | 29.49 |
| **EDMA** | **63.83** | **59.42** | **54.48** | **49.26** | **43.84** | **38.15** | **33.12** | **28.47** |

Table 5: Explanation Fidelity$^+$ (the higher the better) for all baseline methods and our proposed EDMA method using ProNet as the backbone on the EC dataset. The best results are highlighted in bold.

| Top-$k$ | 80 | 90 | 100 | 110 | 120 | 130 | 140 | 150 | 160 |
|---|---|---|---|---|---|---|---|---|---|
| GNNExplainer-Dense | 24.32 | 26.35 | 29.13 | 30.48 | 35.96 | 36.71 | 43.09 | 46.70 | 49.70 |
| PGExplainer-Dense | 26.95 | 24.02 | 27.18 | 30.41 | 28.53 | 30.41 | 31.61 | 37.76 | 40.92 |
| LRI-Bernoulli | 12.09 | 15.77 | 19.37 | 21.77 | 24.55 | 26.58 | 30.48 | 32.21 | 34.08 |
| **EDMA** | **30.41** | **32.28** | **36.94** | **40.09** | **42.34** | **46.40** | **49.92** | **52.48** | **56.38** |

from the other tables due to the special readout function for property $\mu$ in PyG's SchNet implementation; see Appendix F for details. Additional results comparing EDMA with GEM (Lin et al., 2021), a representative baseline based on discrete selection rather than soft-mask optimization, are provided in Appendix G. EDMA achieves higher fidelity across most budgets.

**Classification Task on the EC Dataset.** Additionally, we conduct experiments on the EC dataset with ProNet (Wang et al., 2023) as the backbone model. The results are presented in Table 5 with budget $k$ defined over the range $[80, 160]$ with a step size of 10. Our EDMA outperforms all the baselines by a large margin. Only our method can faithfully explain more than half of the graphs in the dataset, which clearly illustrates the superiority of our proposed method in generalizing to larger and more complex structures.

### 4.3 Qualitative Results

In this section, we present qualitative results of the baselines and our approach in finding the important substructures that contribute to the properties of the molecules the most. Functional groups play a significant role in determining the chemical properties of molecules; thus, an explanation method that generates results with high fidelity is more likely to accurately identify these functional groups. For each molecule, we match the number of atoms identified by the explanation methods to those in the actual chemical explanations across all methods for a fair comparison. Since functional groups are not necessarily connected, we do not impose a requirement for the explanatory subgraphs to be connected. In Fig. 4, we visualize the identified substructures from various explanation methods on several real molecules from the QM9 dataset, focusing on the property $\mu$ with DimeNet++ as the backbone. For clarity, for all explanation methods, the selected atoms within the budget are highlighted uniformly without distinguishing different functional groups, while the last column uses distinct colors to differentiate the contributing functional groups. Clearly, EDMA is effective in identifying explanatory substructures that align closely with established chemical knowledge,

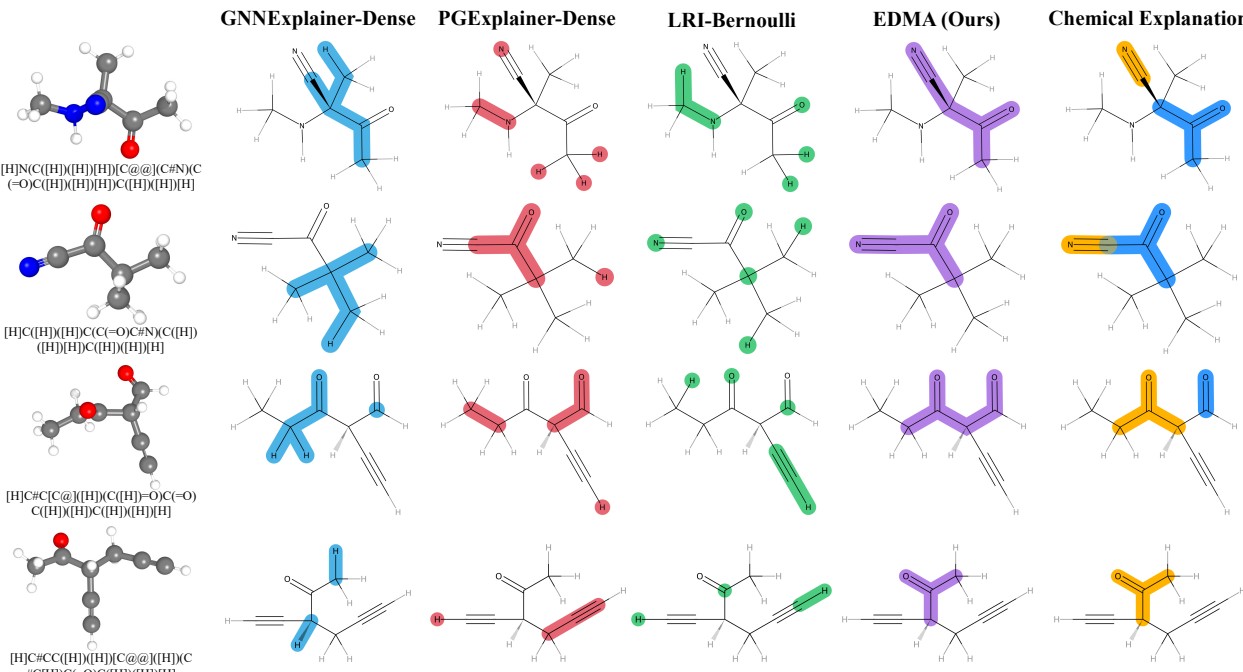

Figure 4: The first column showcases 3D molecules from the QM9 dataset, along with corresponding SMILES strings. The following columns present the explanation results from various baseline methods alongside our EDMA method. Finally, the last column provides chemical interpretations, where distinct colors are used to differentiate the contributing functional groups. **It is evident that our method, EDMA, delivers the most accurate explanation results, aligning closely with chemical priors.**

providing more meaningful and interpretable explanations. In contrast, other baseline methods rely on softer masks, which often lead to less precise and less reliable explanations. By adopting approximately discrete masks, EDMA ensures a higher fidelity to chemical understanding, resulting in more scientifically coherent and robust explanations. Additional results and brief chemical justifications are provided in Appendix H.

## 4.4 Ablation Study

We conduct ablation studies to investigate three fundamental questions: (1) Is the discreteness of masks the primary factor driving improvements in explanation results for 3D graphs, or does the formulation of EBM also contribute? (2) Is there a clear need for methods that *achieve discreteness by design*, such as our proposed EDMA, over regularization-based approaches? (3) Is the outperformance of our proposed method consistent when using equivariant models as backbones?

To address the first question, we isolate the EBM formulation by reducing the effect of discrete masks in our proposed EDMA pipeline. Specifically, we keep all components unchanged except for the parameters related to energy adjustments. In particular, we use a larger $T$, which diminishes the amplification of the energy difference discussed in Sec. 3.3, thereby resulting in relatively less "discrete masks". We refer to this variant as EDMA-soft. The results are presented in Table 6. Clearly, without adjusting the temperature to enforce discreteness, the explanation fidelity of EDMA-soft is much worse than that of EDMA, suggesting that the discreteness of masks is the primary factor in improving explanation fidelity. A theoretical justification for the role of $T$ is provided in Appendix B.1, where we show that the soft-to-discrete gap is explicitly bounded in terms of $T$ and becomes tighter as $T$ decreases.

To address the second question, we remove the regularization term (cross-entropy loss) in Eq. (8) from the baseline models. If the inclusion of such a regularization term is effective, its removal should result in a significant performance downgrade. However, as presented in Table 7, excluding the regularization term from the optimization objective does not lead to any noticeable decline in explanation fidelity; in fact, in

Table 6: Experimental results comparing explanation Fidelity$^-$ of the EDMA method and its soft mask variant, EDMA-soft (achieved by adjusting the temperature hyper-parameter), are presented for the property $\mu$ (dipole moment, in Debye (D)) using SchNet as the backbone model.

| Top-$k$ | 2 | 3 | 4 | 5 | 6 | 7 | 8 | 9 |
|---|---|---|---|---|---|---|---|---|
| EDMA | 2.74 | 3.73 | 4.31 | 4.83 | 5.08 | 5.47 | 5.72 | 5.31 |
| EDMA-soft | 3.32 | 4.52 | 6.14 | 7.20 | 7.88 | 8.09 | 8.02 | 7.44 |

Table 7: Explanation Fidelity$^-$ for all baseline methods with and without (marked with $^*$) entropy loss and our proposed EDMA method using SchNet as the backbone on the property $\mu$ (dipole moment, in Debye (D)). The best results are highlighted in bold.

| Top-$k$ | 2 | 3 | 4 | 5 | 6 | 7 | 8 | 9 |
|---|---|---|---|---|---|---|---|---|
| GNNExplainer-Dense | 3.88 | 5.62 | 7.28 | 8.05 | 8.27 | 8.00 | 7.59 | 6.87 |
| GNNExplainer-Dense$^*$ | 3.82 | 5.90 | 7.40 | 8.11 | 8.09 | 7.83 | 7.44 | 6.75 |
| PGExplainer-Dense | 2.91 | 3.73 | 4.83 | 6.09 | 6.62 | 6.55 | 6.81 | 6.08 |
| PGExplainer-Dense$^*$ | 2.84 | **3.61** | 4.86 | 6.16 | 6.73 | 7.05 | 6.74 | 6.22 |
| LRI-Bernoulli | 3.50 | 4.84 | 6.16 | 6.88 | 7.10 | 7.29 | 7.43 | 7.32 |
| LRI-Bernoulli$^*$ | 3.70 | 5.28 | 6.46 | 6.95 | 7.34 | 7.40 | 7.49 | 7.22 |
| **EDMA** | **2.74** | 3.73 | **4.31** | **4.83** | **5.08** | **5.47** | **5.72** | **5.31** |

Table 8: Explanation Fidelity$^-$ (the lower the better) for GNNExplainer and our proposed EDMA method using EGNN as the backbone on the property $G_f$ (free energy, in eV). The better results are highlighted in bold.

| Top-$k$ | 2 | 3 | 4 | 5 | 6 | 7 | 8 | 9 |
|---|---|---|---|---|---|---|---|---|
| GNNExplainer-Dense | 52.18 | 72.29 | 92.91 | 114.02 | 135.42 | 157.03 | 178.78 | 200.55 |
| **EDMA** | **46.45** | **65.15** | **84.79** | **104.98** | **125.56** | **146.62** | **166.85** | **187.44** |

some cases, it even outperforms the version with regularization. This is caused by the complexities in the optimization process, as discussed in Sec. 3.4. On the other hand, our proposed EDMA method consistently outperforms the baselines, as it achieves discreteness by design rather than by introducing regularization terms that complicate the optimization objective.

To address the third question, we compare the performance of EDMA with GNNExplainer using an equivariant architecture, EGNN (Satorras et al., 2021), on property $G_f$ of the QM9 dataset. The results are presented in Table 8. Our method consistently outperforms GNNExplainer, demonstrating its effectiveness with equivariant models. A brief overview of invariant and equivariant GNNs is provided in Appendix I.

## 5   Conclusions and Future Work

In conclusion, we introduce a principled approach specifically designed to explain 3D GNNs. By recognizing the varying assumptions in 3D GNNs, we identify a common bottleneck shared by existing methods for 3D graph explanations: the gap between the optimized soft masks and the final explanatory substructures due to the non-discreteness of masks. To address this, we propose a novel energy-based explanation function that generates, by pushing up and down the energies, masks that are approximately discrete and highly confident, leading to improved explanation fidelity. Experimental results on both small and large molecules demonstrate the effectiveness and generalizability of our method.

**Future Work.** Our method represents an early effort in 3D molecular graph explanation. It would be interesting to explore whether incorporating additional chemical domain knowledge could further enhance explanation fidelity and generalizability. Moreover, it is worth investigating applying our findings to more complex molecular tasks, such as utilizing the property-informed subgraphs extracted by EDMA for drug editing and molecular optimization tasks, to enable more efficient drug discovery.

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

# A   Theorem Proofs

In this section, we provide a formal theoretical analysis for the soft-to-discrete explanation gap on 3D molecular graphs. Beyond the decomposition in Eq. (5), we derive explicit upper bounds on the discrepancy term under MAE. These bounds depend on (i) the soft-to-hard distance of the node mask and (ii) the graph density (maximum neighborhood size), both of which are defined precisely in the following subsections.

## A.1   MAE-based bound decomposition

**3D graph and masked prediction.** Let $G = (\mathcal{V}, \boldsymbol{X}, \mathbf{A}, \mathbf{R})$ be a 3D molecular graph with $n = |\mathcal{V}|$ nodes, where $\mathbf{R} = [\mathbf{r}_1, \dots, \mathbf{r}_n]^\top \in \mathbb{R}^{n \times 3}$ denotes atom coordinates and $\mathbf{A} \in \{0,1\}^{n \times n}$ is the adjacency matrix (with entries $a_{ij}$). For 3D graphs, we assume $\mathbf{A}$ is constructed using a cutoff radius $r_c$, i.e.,

$$a_{ij} = 1 \iff r_{ij} = \|\mathbf{r}_i - \mathbf{r}_j\|_2 \leq r_c. \tag{A.1}$$

We denote the edge set by $E := \{(i,j) \mid a_{ij} = 1\}$ (treating undirected edges as two directed pairs). A 3D GNN $\Phi$ maps the graph $G$ to a prediction $\hat{Y} \in \mathbb{R}$ for target property $Y$. For any soft edge mask $\tilde{\mathbf{M}} \in [0,1]^{n \times n}$ and the element-wise (Hadamard) product $\odot$, we define the masked prediction

$$\hat{Y}(\tilde{\mathbf{M}}) := \Phi(\mathcal{V}, \boldsymbol{X}, \tilde{\mathbf{M}} \odot \mathbf{A}, \mathbf{R}). \tag{A.2}$$

In particular, we distinguish between a discrete edge mask matrix $\mathbf{M} \in \{0,1\}^{n \times n}$ and a relaxed edge mask matrix $\mathbf{M}' \in [0,1]^{n \times n}$. Following the node-mask parameterization in Eq. (4), we represent the relaxed mask by a soft node mask $\boldsymbol{m} \in [0,1]^n$, which induces the edge mask as:

$$\mathbf{M}' = \boldsymbol{m} \otimes \boldsymbol{m}, \qquad (\mathbf{M}')_{ij} = m_i m_j. \tag{A.3}$$

Similarly, a discretized node mask $\boldsymbol{m}^d \in \{0,1\}^n$ induces a discrete edge mask $\mathbf{M} = \boldsymbol{m}^d \otimes \boldsymbol{m}^d$. Here, $\otimes$ denotes the outer product.

**Loss and discrepancy.** We use the MAE loss $\mathcal{L}(Y; \hat{Y}) := |Y - \hat{Y}|$ and define the prediction discrepancy $d(\hat{Y}_1, \hat{Y}_2) := |\hat{Y}_1 - \hat{Y}_2|$. MAE is 1-Lipschitz in the prediction $\hat{Y}$. We are interested in bounding the discrepancy between $\hat{Y}(\mathbf{M})$ and $\hat{Y}(\mathbf{M}')$.

**Lemma A.1** (Upper bound under MAE). *For any $\mathbf{M} \in \{0,1\}^{n \times n}$ and any $\mathbf{M}' \in [0,1]^{n \times n}$,*

$$\mathcal{L}\Big(Y; \hat{Y}(\mathbf{M})\Big) \leq \mathcal{L}\Big(Y; \hat{Y}(\mathbf{M}')\Big) + d\Big(\hat{Y}(\mathbf{M}'), \hat{Y}(\mathbf{M})\Big). \tag{A.4}$$

*Proof.* By definition of MAE and the triangle inequality on $\mathbb{R}$,

$$\begin{aligned}
\mathcal{L}\Big(Y; \hat{Y}(\mathbf{M})\Big) &= |Y - \hat{Y}(\mathbf{M})| \\
&= \big|(Y - \hat{Y}(\mathbf{M}')) + (\hat{Y}(\mathbf{M}') - \hat{Y}(\mathbf{M}))\big| \\
&\leq |Y - \hat{Y}(\mathbf{M}')| + |\hat{Y}(\mathbf{M}') - \hat{Y}(\mathbf{M})| \\
&= \mathcal{L}\Big(Y; \hat{Y}(\mathbf{M}')\Big) + d\Big(\hat{Y}(\mathbf{M}'), \hat{Y}(\mathbf{M})\Big).
\end{aligned}$$

$\square$

## A.2   Lipschitz Continuity of 3D GNNs w.r.t. Masks

In this section, we provide a formal analysis showing that a broad class of 3D GNN models (including SchNet and DimeNet-type architectures) is Lipschitz continuous with respect to the (edge) explanation mask while holding geometry fixed, under mild assumptions on the network blocks. This result will be used to derive an upper bound for the discrepancy term in Eq. (5).

We consider a broad family of 3D GNN layers that covers continuous-filter message passing and masked message-passing updates. Let $h_i^{(\ell)} \in \mathbb{R}^{d_h}$ denote the representation at layer $\ell$ for node $i$. A generic masked update takes the form

$$h_i^{(\ell+1)}(\tilde{\mathbf{M}}) = U_\ell\Big(h_i^{(\ell)}(\tilde{\mathbf{M}}), s_i^{(\ell)}(\tilde{\mathbf{M}})\Big), \qquad s_i^{(\ell)}(\tilde{\mathbf{M}}) = \sum_{j \in \mathcal{N}(i)} \tilde{\mathbf{M}}_{ij}\, \Psi_\ell\Big(h_i^{(\ell)}(\tilde{\mathbf{M}}), h_j^{(\ell)}(\tilde{\mathbf{M}}), \mathbf{e}_{ij}\Big), \qquad \text{(A.5)}$$

where $\mathbf{e}_{ij}$ are geometry-derived edge features computed from the fixed coordinates (e.g., a radial basis expansion of $r_{ij} = \|\mathbf{r}_i - \mathbf{r}_j\|_2$), and $U_\ell$ and $\Psi_\ell$ are neural blocks (e.g., linear/MLP layers with Lipschitz activation functions). The neighborhood $\mathcal{N}(i)$ is induced by the cutoff adjacency matrix $\mathbf{A}$.

Drawing from the principle of atomic energy decomposition in quantum chemistry, it is chemically intuitive to approximate the total molecular property as a sum of local atomic contributions. Consequently, 3D GNNs typically use a permutation-invariant readout to map node representations to a graph-level prediction:

$$\hat{Y}(\tilde{\mathbf{M}}) = R\Big(H^{(L)}(\tilde{\mathbf{M}})\Big), \qquad H^{(L)}(\tilde{\mathbf{M}}) := \{h_i^{(L)}(\tilde{\mathbf{M}})\}_{i=1}^n, \qquad \text{(A.6)}$$

where $R$ is a permutation-invariant pooling (e.g., sum/mean) followed by an MLP for property prediction.

We first state the conditions and assumptions used in the proof. Recall from Appendix A.1 that $E := \{(i,j) \mid a_{ij} = 1\}$ denotes the (ordered) edge set induced by $\mathbf{A}$.

**Condition A.2** (Finite cutoff neighborhoods). Edges are constructed by a cutoff radius $r_c$, i.e., $a_{ij} = 1 \iff r_{ij} = \|\mathbf{r}_i - \mathbf{r}_j\|_2 \le r_c$. Thus each node has a finite neighborhood $\mathcal{N}(i) = \{j \mid a_{ij} = 1\}$, and we define $d_{\max} := \max_i |\mathcal{N}(i)| < \infty$.

**Condition A.3** (Bounded geometric features). Let $\mathbf{e}_{ij}$ denote geometry-derived edge features computed from distances $r_{ij} \in [0, r_c]$ (and, for DimeNet-type models, angles in $[0, \pi]$) via continuous basis expansions. Then $\mathbf{e}_{ij}$ is bounded for all $(i,j) \in E$, since continuous functions attain maxima on compact sets.

**Assumption A.4** (Bounded/Lipschitz message function). For each layer $\ell$, there exist constants $B_{\Psi_\ell}, L_{\Psi_\ell} > 0$ such that for all admissible edge features $\mathbf{e}_{ij}$ and for all node states $h_i, h_j, h_i', h_j'$ encountered on the data domain,

$$\|\Psi_\ell(h_i, h_j, \mathbf{e}_{ij})\|_2 \le B_{\Psi_\ell}, \qquad \text{(A.7)}$$

$$\|\Psi_\ell(h_i, h_j, \mathbf{e}_{ij}) - \Psi_\ell(h_i', h_j', \mathbf{e}_{ij})\|_2 \le L_{\Psi_\ell}\big(\|h_i - h_i'\|_2 + \|h_j - h_j'\|_2\big). \qquad \text{(A.8)}$$

**Assumption A.5** (Lipschitz update function). For each layer $\ell$, the update map $U_\ell : \mathbb{R}^{d_h} \times \mathbb{R}^{d_h} \to \mathbb{R}^{d_h}$ is Lipschitz in both arguments: there exists $L_{U_\ell} > 0$ such that for all $a, a', b, b'$,

$$\|U_\ell(a, b) - U_\ell(a', b')\|_2 \le L_{U_\ell}\big(\|a - a'\|_2 + \|b - b'\|_2\big). \qquad \text{(A.9)}$$

**Assumption A.6** (Lipschitz readout). There exists $L_R > 0$ such that the readout function $R$ satisfies, for any two node-representation sets $H = \{h_i\}_{i=1}^n$ and $H' = \{h_i'\}_{i=1}^n$,

$$|R(H) - R(H')| \le L_R \|H - H'\|_{1,V}. \qquad \text{(A.10)}$$

*Norm conventions.* We use $\|\cdot\|_2$ for the vector $\ell_2$ norm. For node representations $H = \{h_i\}_{i=1}^n$ with $h_i \in \mathbb{R}^{d_h}$, we define the mixed norm

$$\|H\|_{1,V} := \sum_{i=1}^n \|h_i\|_2, \qquad \|H - H'\|_{1,V} := \sum_{i=1}^n \|h_i - h_i'\|_2. \qquad \text{(A.11)}$$

For edge masks $\tilde{\mathbf{M}} \in \mathbb{R}^{n \times n}$, we define the edge-wise $\ell_1$ norm restricted to edges

$$\|\tilde{\mathbf{M}}\|_{1,E} := \sum_{(i,j) \in E} |\tilde{\mathbf{M}}_{ij}|, \qquad \|(\tilde{\mathbf{M}} - \tilde{\mathbf{M}}') \odot \mathbf{A}\|_{1,E} := \sum_{(i,j) \in E} |\tilde{\mathbf{M}}_{ij} - \tilde{\mathbf{M}}_{ij}'|. \qquad \text{(A.12)}$$

**Remark.** Assumptions A.4, A.5, and A.6 are standard for networks built from linear layers/MLPs with finite operator norms and Lipschitz activations (e.g., ReLU, tanh), and hold on compact (bounded) domains.

**Theorem A.7** (Lipschitz continuity in the edge mask). *Under Conditions A.2–A.3 and Assumptions A.4–A.6, there exists a constant $L_\Phi < \infty$ such that for any two masks $\tilde{\mathbf{M}}, \tilde{\mathbf{M}}' \in [0,1]^{n \times n}$,*

$$\left| \hat{Y}(\tilde{\mathbf{M}}) - \hat{Y}(\tilde{\mathbf{M}}') \right| \leq L_\Phi \, \|(\tilde{\mathbf{M}} - \tilde{\mathbf{M}}') \odot \mathbf{A}\|_{1,E}. \tag{A.13}$$

*Proof.* Let $\Delta h_i^{(\ell)} := h_i^{(\ell)}(\tilde{\mathbf{M}}) - h_i^{(\ell)}(\tilde{\mathbf{M}}')$ and $\Delta H^{(\ell)} := \{\Delta h_i^{(\ell)}\}_{i=1}^n$. Also define $\Delta s_i^{(\ell)} := s_i^{(\ell)}(\tilde{\mathbf{M}}) - s_i^{(\ell)}(\tilde{\mathbf{M}}')$. By Assumption A.5,

$$\|\Delta h_i^{(\ell+1)}\|_2 \leq L_{U_\ell} \Big( \|\Delta h_i^{(\ell)}\|_2 + \|\Delta s_i^{(\ell)}\|_2 \Big). \tag{A.14}$$

Using the definition of $s_i^{(\ell)}(\cdot)$ in Eq. (A.5), we decompose

$$\begin{aligned}
\Delta s_i^{(\ell)} &= \sum_{j \in \mathcal{N}(i)} (\tilde{\mathbf{M}}_{ij} - \tilde{\mathbf{M}}'_{ij}) \Psi_\ell \Big( h_i^{(\ell)}(\tilde{\mathbf{M}}'), h_j^{(\ell)}(\tilde{\mathbf{M}}'), \mathbf{e}_{ij} \Big) \\
&+ \sum_{j \in \mathcal{N}(i)} \tilde{\mathbf{M}}_{ij} \Big[ \Psi_\ell \Big( h_i^{(\ell)}(\tilde{\mathbf{M}}), h_j^{(\ell)}(\tilde{\mathbf{M}}), \mathbf{e}_{ij} \Big) - \Psi_\ell \Big( h_i^{(\ell)}(\tilde{\mathbf{M}}'), h_j^{(\ell)}(\tilde{\mathbf{M}}'), \mathbf{e}_{ij} \Big) \Big].
\end{aligned} \tag{A.15}$$

Taking norms and applying the triangle inequality, together with (i) boundedness of $\Psi_\ell$ (Eq. (A.7)), (ii) $\tilde{\mathbf{M}}_{ij}, \tilde{\mathbf{M}}'_{ij} \in [0,1]$, and (iii) Lipschitzness of $\Psi_\ell$ (Eq. (A.8)), we obtain

$$\|\Delta s_i^{(\ell)}\|_2 \leq B_{\Psi_\ell} \sum_{j \in \mathcal{N}(i)} |\tilde{\mathbf{M}}_{ij} - \tilde{\mathbf{M}}'_{ij}| + L_{\Psi_\ell} \sum_{j \in \mathcal{N}(i)} \Big( \|\Delta h_i^{(\ell)}\|_2 + \|\Delta h_j^{(\ell)}\|_2 \Big). \tag{A.16}$$

Summing Eq. (A.14) over $i$ and substituting Eq. (A.16) yields

$$\begin{aligned}
\|\Delta H^{(\ell+1)}\|_{1,V} &= \sum_{i=1}^n \|\Delta h_i^{(\ell+1)}\|_2 \\
&\leq L_{U_\ell} \|\Delta H^{(\ell)}\|_{1,V} + L_{U_\ell} \sum_{i=1}^n \|\Delta s_i^{(\ell)}\|_2 \\
&\leq L_{U_\ell} \|\Delta H^{(\ell)}\|_{1,V} + L_{U_\ell} B_{\Psi_\ell} \sum_{i=1}^n \sum_{j \in \mathcal{N}(i)} |\tilde{\mathbf{M}}_{ij} - \tilde{\mathbf{M}}'_{ij}| \\
&+ L_{U_\ell} L_{\Psi_\ell} \sum_{i=1}^n \sum_{j \in \mathcal{N}(i)} \Big( \|\Delta h_i^{(\ell)}\|_2 + \|\Delta h_j^{(\ell)}\|_2 \Big).
\end{aligned} \tag{A.17}$$

For the mask term, by the definition of $\|\cdot\|_{1,E}$,

$$\sum_{i=1}^n \sum_{j \in \mathcal{N}(i)} |\tilde{\mathbf{M}}_{ij} - \tilde{\mathbf{M}}'_{ij}| = \|(\tilde{\mathbf{M}} - \tilde{\mathbf{M}}') \odot \mathbf{A}\|_{1,E}. \tag{A.18}$$

For the state terms, using $|\mathcal{N}(i)| \leq d_{\max}$ from Condition A.2,

$$\sum_{i=1}^n \sum_{j \in \mathcal{N}(i)} \|\Delta h_i^{(\ell)}\|_2 \leq d_{\max} \sum_{i=1}^n \|\Delta h_i^{(\ell)}\|_2 = d_{\max} \|\Delta H^{(\ell)}\|_{1,V}, \tag{A.19}$$

and similarly,

$$\sum_{i=1}^n \sum_{j \in \mathcal{N}(i)} \|\Delta h_j^{(\ell)}\|_2 \leq d_{\max} \|\Delta H^{(\ell)}\|_{1,V}. \tag{A.20}$$

Therefore Eq. (A.17) simplifies to

$$\|\Delta H^{(\ell+1)}\|_{1,V} \leq a_\ell \|\Delta H^{(\ell)}\|_{1,V} + b_\ell \|(\tilde{\mathbf{M}} - \tilde{\mathbf{M}}') \odot \mathbf{A}\|_{1,E}, \tag{A.21}$$

where

$$a_\ell := L_{U_\ell}\big(1 + 2d_{\max}L_{\Psi_\ell}\big), \qquad b_\ell := L_{U_\ell}B_{\Psi_\ell}. \tag{A.22}$$

We note that $\Delta H^{(0)} = 0$ since the layer-0 embeddings $h_i^{(0)}$ depend only on the input features (and geometry) and are independent of the edge mask; masks are applied only to message passing. Unrolling Eq. (A.21) across layers $\ell = 0, \dots, L-1$ yields

$$\|\Delta H^{(L)}\|_{1,V} \le \left(\sum_{t=0}^{L-1} b_t \prod_{s=t+1}^{L-1} a_s\right) \|(\tilde{\mathbf{M}} - \tilde{\mathbf{M}}') \odot \mathbf{A}\|_{1,E}. \tag{A.23}$$

Finally, by Assumption A.6,

$$|\hat{Y}(\tilde{\mathbf{M}}) - \hat{Y}(\tilde{\mathbf{M}}')| = |R(H^{(L)}(\tilde{\mathbf{M}})) - R(H^{(L)}(\tilde{\mathbf{M}}'))| \le L_R\|\Delta H^{(L)}\|_{1,V}. \tag{A.24}$$

Combining the above inequalities establishes Eq. (A.13) with

$$L_\Phi := L_R \sum_{t=0}^{L-1} b_t \prod_{s=t+1}^{L-1} a_s < \infty. \tag{A.25}$$

$\square$

**Instantiation to SchNet and DimeNet(++).** **SchNet.** SchNet employs cutoff-based neighborhoods (Condition A.2) and continuous-filter message passing, where the message function $\Psi_\ell(\cdot, \cdot, \mathbf{e}_{ij})$ is computed from a radial basis expansion of $r_{ij} \in [0, r_c]$ (hence $\mathbf{e}_{ij}$ is bounded by Condition A.3) followed by MLP blocks. Therefore SchNet admits the masked update form in Eq. (A.5). Moreover, for networks composed of linear/MLP layers with Lipschitz activations and bounded hidden states on the data domain, Assumptions A.4, A.5, and A.6 hold. Hence Theorem A.7 applies, and SchNet is Lipschitz continuous with respect to the edge mask $\tilde{\mathbf{M}}$ (with geometry fixed).

**DimeNet/DimeNet++.** DimeNet and DimeNet++ additionally incorporate angle-dependent (triplet) interactions. These terms can be expressed as finite sums over triplets $(i, j, k)$ induced by the cutoff graph and computed from bounded radial/angular basis features with $r_{ij} \in [0, r_c]$ and $\angle jik \in [0, \pi]$ (Condition A.3). In typical masked variants, a triplet contribution is multiplied by a bounded mask weight that depends on the involved pairs (e.g., $\tilde{\mathbf{M}}_{ij}$, or $\tilde{\mathbf{M}}_{ij}\tilde{\mathbf{M}}_{ik}$); since $\tilde{\mathbf{M}} \in [0,1]^{n\times n}$, this mask factor is itself Lipschitz in $\tilde{\mathbf{M}}$ under the $\|\cdot\|_{1,E}$ norm. Consequently, the resulting update remains a finite sum of bounded/Lipschitz components, and the same layerwise argument extends with Lipschitz constants depending on an upper bound on the number of triplets per node. In the worst case this count scales as $O(d_{\max}^2)$, yielding a correspondingly larger (but still finite) constant $L_\Phi$, and Lipschitz continuity with respect to $\tilde{\mathbf{M}}$ also holds.

## A.3 Discrepancy Bound for Soft vs. Discrete Node Masks

In this section, we combine the MAE decomposition (Appendix A.1) with the Lipschitz continuity of 3D GNNs with respect to edge masks (Appendix A.2) to obtain an explicit upper bound on the discrepancy between the optimized soft mask and the final discrete explanation. The discrepancy term in Eq. (5) can be upper bounded using Theorem A.7. Under Conditions A.2–A.3 and Assumptions A.4–A.6, for any $\mathbf{M} \in \{0,1\}^{n\times n}$ and $\mathbf{M}' \in [0,1]^{n\times n}$,

$$|\hat{Y}(\mathbf{M}') - \hat{Y}(\mathbf{M})| \le L_\Phi \, \|(\mathbf{M}' - \mathbf{M}) \odot \mathbf{A}\|_{1,E}, \tag{A.26}$$

where $L_\Phi$ is the Lipschitz constant in Theorem A.7.

We now exploit the outer-product construction $\mathbf{M}' = \boldsymbol{m} \otimes \boldsymbol{m}$ to relate edge-mask discrepancy to the distance between node masks.

**Lemma A.8** (Outer-product mask discrepancy). *Let $\boldsymbol{m} \in [0,1]^n$ and $\boldsymbol{m}^d \in \{0,1\}^n$. Define $\mathbf{M}' = \boldsymbol{m} \otimes \boldsymbol{m}$ and $\mathbf{M} = \boldsymbol{m}^d \otimes \boldsymbol{m}^d$. Let $\deg(i) := |\mathcal{N}(i)|$, where $\mathcal{N}(i) = \{j : (i,j) \in E\}$ is the cutoff neighborhood. Then*

$$\|(\mathbf{M}' - \mathbf{M}) \odot \mathbf{A}\|_{1,E} \le 2 \sum_{i=1}^{n} \deg(i) \, |m_i - m_i^d|. \tag{A.27}$$

*Consequently,*

$$\|(\mathbf{M}' - \mathbf{M}) \odot \mathbf{A}\|_{1,E} \le 2 d_{\max} \|\boldsymbol{m} - \boldsymbol{m}^d\|_1. \tag{A.28}$$

*Proof.* For any edge $(i,j) \in E$,

$$\begin{aligned}
|m_i m_j - m_i^d m_j^d| &= |(m_i - m_i^d)m_j + m_i^d(m_j - m_j^d)| \\
&\le |m_i - m_i^d| \, |m_j| + |m_i^d| \, |m_j - m_j^d| \\
&\le |m_i - m_i^d| + |m_j - m_j^d|,
\end{aligned} \tag{A.29}$$

since $m_j \in [0,1]$ and $m_i^d \in \{0,1\} \subset [0,1]$. Summing Eq. (A.29) over edges gives

$$\begin{aligned}
\sum_{(i,j) \in E} |m_i m_j - m_i^d m_j^d| &\le \sum_{(i,j) \in E} \left( |m_i - m_i^d| + |m_j - m_j^d| \right) \\
&= 2 \sum_{i=1}^{n} \deg(i) \, |m_i - m_i^d|,
\end{aligned} \tag{A.30}$$

which proves Eq. (A.27). Finally,

$$2 \sum_{i=1}^{n} \deg(i) \, |m_i - m_i^d| \le 2 d_{\max} \sum_{i=1}^{n} |m_i - m_i^d| = 2 d_{\max} \|\boldsymbol{m} - \boldsymbol{m}^d\|_1, \tag{A.31}$$

proving Eq. (A.28). $\qquad\square$

**Remark.** Eq. (A.27) and (A.28) show that the soft-to-discrete discrepancy is controlled by the *soft-to-hard distance* of the node mask, $\|\boldsymbol{m} - \boldsymbol{m}^d\|_1$, and is amplified by graph density through $\deg(i)$ (or $d_{\max}$ in the worst case).

- **Worst case (non-binary soft masks).** If many nodes have mask values not close to 0 or 1, then $|m_i - m_i^d|$ can be large for many $i$, and the bound grows proportionally to $\sum_i \deg(i)|m_i - m_i^d|$ (or to $d_{\max}\|\boldsymbol{m} - \boldsymbol{m}^d\|_1$ in the worst case).
- **Effect of confidence/discreteness.** Encouraging $m_i$ to concentrate near $\{0,1\}$ reduces $|m_i - m_i^d|$ for most nodes, directly tightening the bound; this is most impactful in dense cutoff graphs, where each such node influences many edges via $\mathbf{M}' = \boldsymbol{m} \otimes \boldsymbol{m}$.

**Theorem A.9** (Soft-to-discrete bound under MAE). *Let $\boldsymbol{m} \in [0,1]^n$ induce $\mathbf{M}' = \boldsymbol{m} \otimes \boldsymbol{m}$ and let $\boldsymbol{m}^d \in \{0,1\}^n$ induce $\mathbf{M} = \boldsymbol{m}^d \otimes \boldsymbol{m}^d$. Under Conditions A.2–A.3 and Assumptions A.4–A.6,*

$$|Y - \hat{Y}(\mathbf{M})| \le |Y - \hat{Y}(\mathbf{M}')| + 2 L_\Phi \sum_{i=1}^{n} \deg(i) \, |m_i - m_i^d| \le |Y - \hat{Y}(\mathbf{M}')| + 2 L_\Phi d_{\max} \|\boldsymbol{m} - \boldsymbol{m}^d\|_1. \tag{A.32}$$

*Proof.* Apply Lemma A.1 with $\mathbf{M}$ and $\mathbf{M}'$ to obtain $|Y - \hat{Y}(\mathbf{M})| \le |Y - \hat{Y}(\mathbf{M}')| + |\hat{Y}(\mathbf{M}') - \hat{Y}(\mathbf{M})|$. Then bound the prediction gap using Theorem A.7 and Lemma A.8. $\qquad\square$

**Remark.** Eq. (A.32) highlights a degree-amplification effect: the prediction gap induced by soft-to-discrete discrepancy is controlled by $L_\Phi$ and increases with graph density (via $d_{\max}$). Therefore, in 3D cutoff graphs—where neighborhoods grow quickly with the cutoff radius—non-binary mask values can be amplified through many induced edges, making explanations less stable unless the method explicitly enforces discreteness.

# B Theoretical Analysis of EDMA and Related Relaxations

This section provides further analysis of why EDMA reduces the soft-to-discrete discrepancy characterized in Appendix A. We analyze EDMA's energy-based parameterization and show that it yields explicit control of the soft-to-hard distance of masks, and we contrast this behavior with common discretization/regularization alternatives used in the literature.

## B.1 Energy gaps induce confident (nearly discrete) masks

For each node $i$, EDMA assigns two scalar energy values $\mathcal{E}_0(\boldsymbol{e}_i), \mathcal{E}_1(\boldsymbol{e}_i) \in \mathbb{R}$ corresponding to two states (e.g., *unimportance* vs. *importance*). We define the energy gap

$$\Delta \mathcal{E}_i := \mathcal{E}_1(\boldsymbol{e}_i) - \mathcal{E}_0(\boldsymbol{e}_i). \tag{B.1}$$

Given a temperature $T > 0$, EDMA defines a soft mask probability

$$m_i(T) := \sigma\left(\frac{\Delta \mathcal{E}_i}{T}\right) \in (0,1), \qquad \sigma(x) = \frac{1}{1 + e^{-x}}. \tag{B.2}$$

The corresponding hard gate is the zero-temperature limit (with arbitrary tie-breaking at $\Delta \mathcal{E}_i = 0$):

$$m_i^d := \mathbb{1}\{\Delta \mathcal{E}_i \geq 0\} \in \{0,1\}. \tag{B.3}$$

The mapping in Eq. (B.2) admits a variational characterization: it is the unique minimizer of an entropy-regularized linear objective, making the role of $T$ explicit.

**Proposition B.1** (Entropy-regularized variational form). *Fix $T > 0$ and a node $i$. Define the negative binary entropy $H(m) := m \log m + (1-m) \log(1-m)$ for $m \in (0,1)$ (with the usual continuous extension to $[0,1]$). Then $m_i(T)$ in Eq. (B.2) is the unique minimizer of*

$$m_i(T) = \arg \min_{m \in (0,1)} \left\{ -\Delta \mathcal{E}_i\, m + T\, H(m) \right\}. \tag{B.4}$$

*Proof.* Consider $J(m) := -\Delta \mathcal{E}_i\, m + T\left(m \log m + (1-m) \log(1-m)\right)$ on $m \in (0,1)$. Since $H$ is strictly convex on $(0,1)$, $J$ is strictly convex and hence has a unique minimizer characterized by the first-order condition

$$0 = J'(m) = -\Delta \mathcal{E}_i + T(\log m - \log(1-m)) = -\Delta \mathcal{E}_i + T \log\left(\frac{m}{1-m}\right). \tag{B.5}$$

Rearranging gives $\frac{m}{1-m} = e^{\Delta \mathcal{E}_i / T}$, hence $m = \sigma(\Delta \mathcal{E}_i / T)$, which is Eq. (B.2). $\square$

**Remark.** Proposition B.1 shows that decreasing $T$ weakens the entropy regularization and drives $m_i(T) = \sigma(\Delta \mathcal{E}_i / T)$ toward a near-binary decision. In particular, for any fixed energy gap $\Delta \mathcal{E}_i \neq 0$, we have $m_i(T) \to m_i^d$ as $T \to 0^+$ (with the tie case $\Delta \mathcal{E}_i = 0$ handled by the chosen convention). Next, we quantify the convergence rate of $m_i(T)$ to the hard decision $m_i^d$ as a function of $|\Delta \mathcal{E}_i|$ and the temperature $T$.

**Lemma B.2** (Energy-gap-to-confidence bound). *Let $m_i(T)$ and $m_i^d$ be defined in Eq. (B.2) and (B.3). Then for every $i$,*

$$|m_i(T) - m_i^d| \leq \exp\left(-\frac{|\Delta \mathcal{E}_i|}{T}\right). \tag{B.6}$$

*Consequently,*

$$\|\boldsymbol{m}(T) - \boldsymbol{m}^d\|_1 = \sum_{i=1}^n |m_i(T) - m_i^d| \leq \sum_{i=1}^n \exp\left(-\frac{|\Delta \mathcal{E}_i|}{T}\right). \tag{B.7}$$

*Proof.* Fix $i$. If $\Delta \mathcal{E}_i \geq 0$, then $m_i^d = 1$ and

$$|m_i(T) - m_i^d| = 1 - \sigma\left(\frac{\Delta \mathcal{E}_i}{T}\right) = \sigma\left(-\frac{\Delta \mathcal{E}_i}{T}\right) = \frac{1}{1 + \exp\left(\frac{\Delta \mathcal{E}_i}{T}\right)} \leq \exp\left(-\frac{\Delta \mathcal{E}_i}{T}\right), \tag{B.8}$$

where the last inequality uses $\frac{1}{1+e^x} \leq e^{-x}$ for all $x \geq 0$. If $\Delta\mathcal{E}_i < 0$, then $m_i^d = 0$ and

$$|m_i(T) - m_i^d| = \sigma\left(\frac{\Delta\mathcal{E}_i}{T}\right) = \frac{1}{1 + \exp\left(\frac{|\Delta\mathcal{E}_i|}{T}\right)} \leq \exp\left(-\frac{|\Delta\mathcal{E}_i|}{T}\right), \tag{B.9}$$

again by $\frac{1}{1+e^x} \leq e^{-x}$ for $x \geq 0$. Combining the two cases yields Eq. (B.6). Summing Eq. (B.6) over $i = 1, \ldots, n$ gives Eq. (B.7). $\qquad\square$

**Remark.** Lemma B.2 shows that EDMA's temperature-scaled energy-gap mapping yields an *exponential* control of the soft-to-hard distance $\|\boldsymbol{m}(T) - \boldsymbol{m}^d\|_1$ in terms of the normalized margins $|\Delta\mathcal{E}_i|/T$. In particular, decreasing $T$ (or increasing energy gaps) rapidly forces $m_i(T)$ toward $\{0, 1\}$, thereby directly tightening the soft-to-discrete discrepancy bound in Appendix A.

We now connect the energy-gap control in Lemma B.2 to the soft-to-discrete prediction gap. Combining Lemma B.2 with the outer-product discrepancy bound (Lemma A.8) gives

$$\big\|(\mathbf{M}'(T) - \mathbf{M}) \odot \mathbf{A}\big\|_{1,E} \leq 2d_{\max} \sum_{i=1}^{n} \exp\left(-\frac{|\Delta\mathcal{E}_i|}{T}\right). \tag{B.10}$$

If the energy gaps are uniformly bounded away from zero, i.e.,

$$|\Delta\mathcal{E}_i| \geq \Delta_{\min} > 0, \qquad \forall i, \tag{B.11}$$

then Eq. (B.10) simplifies to

$$\big\|(\mathbf{M}'(T) - \mathbf{M}) \odot \mathbf{A}\big\|_{1,E} \leq 2d_{\max} n \exp\left(-\frac{\Delta_{\min}}{T}\right). \tag{B.12}$$

Finally, applying the Lipschitz continuity of the predictor w.r.t. edge masks (Theorem A.7) yields

$$\big|\hat{Y}(\mathbf{M}'(T)) - \hat{Y}(\mathbf{M})\big| \leq 2L_\Phi d_{\max} \sum_{i=1}^{n} \exp\left(-\frac{|\Delta\mathcal{E}_i|}{T}\right), \tag{B.13}$$

and under Eq. (B.11),

$$\big|\hat{Y}(\mathbf{M}'(T)) - \hat{Y}(\mathbf{M})\big| \leq 2L_\Phi d_{\max} n \exp\left(-\frac{\Delta_{\min}}{T}\right). \tag{B.14}$$

Eq. (B.13) and (B.14) make the role of EDMA explicit: larger energy gaps and/or smaller temperature $T$ drive $\mathbf{M}'(T)$ exponentially close to the discrete mask $\mathbf{M}$, thereby shrinking the soft-to-discrete prediction gap. The factor $d_{\max}$ highlights the density amplification in 3D cutoff graphs, motivating confidence/discreteness enforcement.

**Remark.** The exponential term in our bound makes the role of the temperature $T$ explicit: for fixed energy gaps $\{|\Delta\mathcal{E}_i|\}_{i=1}^{n}$, the quantity $\sum_{i=1}^{n} \exp(-|\Delta\mathcal{E}_i|/T)$ is monotonically increasing in $T$. Thus, a larger temperature produces less-confident (more continuous) masks and yields a looser worst-case soft-to-discrete discrepancy bound, while annealing $T$ tightens the bound by driving $m_i(T)$ closer to $\{0, 1\}$. This directly explains why the EDMA-soft variant (using a larger $T$) can underperform EDMA in explanation fidelity in Sec. 4.4.

## B.2 Comparison to other discretization relaxations

This subsection contrasts EDMA with three common alternatives for obtaining (approximately) discrete masks: (i) direct $\ell_0$ regularization surrogates, (ii) hard-concrete / $\ell_0$-style gates, and (iii) Gumbel-Softmax / Concrete relaxations. The goal is not to claim these methods cannot work, but to clarify *what they guarantee* and *which failure modes remain* for dense 3D graphs.

### B.2.1 Direct $\ell_0$ regularization (deterministic or surrogate)

Many approaches add a sparsity penalty such as $\lambda\|\boldsymbol{m}\|_0$ (or a surrogate $\lambda\sum_i m_i$, $\lambda\sum_i \text{HardSigmoid}(m_i)$, etc.) while optimizing a soft mask $\boldsymbol{m} \in [0,1]^n$.

$\ell_0$ **promotes sparsity but not necessarily discreteness.** A sparsity penalty primarily encourages many $m_i$ to be small, but it does not by itself prevent the remaining active nodes from taking intermediate values. In particular, when optimizing a soft node mask $\boldsymbol{m} \in [0,1]^n$ (and the induced edge mask $\mathbf{M}'(\boldsymbol{m}) = \boldsymbol{m} \otimes \boldsymbol{m}$), stationary points can occur with $m_i \in (0,1)$ due to a trade-off between the prediction term and the regularizer.

To formalize this, consider a generic differentiable sparsity surrogate $p : [0,1] \to \mathbb{R}$ applied elementwise, e.g., $p(m_i) = m_i$ or a smooth approximation of the step/counting function. Consider the objective

$$J(\boldsymbol{m}) = \left|Y - \hat{Y}(\mathbf{M}'(\boldsymbol{m}))\right| + \lambda \sum_{i=1}^{n} p(m_i), \qquad \lambda > 0, \tag{B.15}$$

where $\mathbf{M}'(\boldsymbol{m}) = \boldsymbol{m} \otimes \boldsymbol{m}$.

At an interior stationary point $m_i \in (0,1)$ (so box constraints are inactive), a first-order necessary condition (using subgradients for the MAE term) takes the form

$$0 \in \partial_{m_i}\left|Y - \hat{Y}(\mathbf{M}'(\boldsymbol{m}))\right| + \lambda\, p'(m_i), \tag{B.16}$$

which can hold with $m_i \in (0,1)$ whenever the prediction subgradient balances the regularizer gradient. Thus, $\ell_0$-style sparsity is compatible with *non-binary* active masks, and therefore can still yield a large soft-to-hard discrepancy in dense 3D graphs.

### B.2.2 Hard-concrete

Hard-concrete (and related $\ell_0$ gates) introduce *stochastic* node gates $z_i \in [0,1]$ during training. A common parameterization is

$$u_i \sim \text{Uniform}(0,1), \qquad s_i = \sigma\left(\frac{\log u_i - \log(1 - u_i) + \log \alpha_i}{T}\right), \qquad z_i = \text{clamp}(s_i(\zeta - \gamma) + \gamma, 0, 1), \tag{B.17}$$

where $(\gamma, \zeta)$ stretch the distribution and $T$ is a temperature. Sparsity is typically encouraged by penalizing $\mathbb{P}(z_i > 0)$ (or a closely related expectation surrogate), yielding an expected-$\ell_0$ style penalty.

For our analysis, however, the key quantity is not sparsity per se but the *soft-to-hard distance* that enters the discrepancy amplification in dense cutoff graphs. Let the induced (random) edge mask be $\mathbf{M}'(\mathbf{z}) := \mathbf{z} \otimes \mathbf{z}$, and let $\mathbf{M} := \boldsymbol{m}^d \otimes \boldsymbol{m}^d$ be a deterministic discrete mask. By Lipschitz continuity in the edge mask (Theorem A.7) and Jensen's inequality,

$$\mathbb{E}\left[\,\left|\hat{Y}(\mathbf{M}'(\mathbf{z})) - \hat{Y}(\mathbf{M})\right|\,\right] \leq L_\Phi\, \mathbb{E}\left[\left\|(\mathbf{M}'(z) - \mathbf{M}) \odot \mathbf{A}\right\|_{1,E}\right]. \tag{B.18}$$

Moreover, since $z_i \in [0,1]$ almost surely, Lemma A.8 applies pointwise to each realization of $z$, and taking expectations yields

$$\mathbb{E}\left[\left\|(\mathbf{M}'(\mathbf{z}) - \mathbf{M}) \odot \mathbf{A}\right\|_{1,E}\right] \leq 2\sum_{i=1}^{n} \deg(i)\, \mathbb{E}[|z_i - m_i^d|] \leq 2d_{\max}\sum_{i=1}^{n} \mathbb{E}[|z_i - m_i^d|]. \tag{B.19}$$

Eq. (B.19) highlights the key limitation for dense 3D graphs: **an expected-$\ell_0$ penalty can make $\mathbb{P}(z_i > 0)$ small, yet it does not by itself guarantee that $z_i$ concentrates near $\{0,1\}$ with high probability.** As a result, intermediate-probability gates can persist during optimization, and their effect is amplified by $d_{\max}$ through the outer product $\mathbf{z} \otimes \mathbf{z}$. In contrast, EDMA yields *deterministic, margin-controlled* near-binary masks under annealing, directly targeting the quantity that governs the soft-to-discrete discrepancy.

### B.2.3 Gumbel-Softmax / Concrete relaxation

Gumbel-Softmax typically produces a differentiable approximation to a discrete choice using Gumbel noise. For a Bernoulli gate, one common form is

$$g_i \sim \text{Gumbel}(0,1), \qquad m_i(T) = \sigma\left(\frac{\theta_i + g_i}{T}\right), \tag{B.20}$$

with a straight-through discretization $m_i^d = \mathbb{1}\{\theta_i + g_i \geq 0\}$.

**Variance and lack of deterministic margin control.** Like hard-concrete, the key difference from EDMA is that Eq. (B.20) injects randomness via $g_i$. The soft-to-hard distance control becomes *distributional* rather than deterministic:

$$|m_i(T) - m_i^d| = \left|\sigma\left(\frac{\theta_i + g_i}{T}\right) - \mathbb{1}\{\theta_i + g_i \geq 0\}\right|, \tag{B.21}$$

whose magnitude depends on the random realization $g_i$ and cannot be bounded by a deterministic function of a learned margin alone unless one conditions on events such as $|\theta_i + g_i| \geq \Delta$ holding with high probability. In dense graphs, such randomness can lead to a larger variance in the induced edge mask $\mathbf{M}'$, and thus greater instability of explanations.

**Remark.** In dense 3D graphs with soft edge masks, the main difficulty is not sparsity per se but *confidence/discreteness*: fractional node probabilities induce many fractional edges and can be amplified by graph density. Standard relaxations (e.g., $\ell_0$-style penalties, hard-concrete, and Gumbel-Softmax) often prioritize sparsity or provide expectation-level control under stochastic gating, which does not directly guarantee a small soft-to-hard distance. EDMA instead targets this quantity explicitly by producing deterministic, margin-controlled near-binary masks under temperature annealing, thereby directly shrinking the soft-to-discrete discrepancy term in Eq. (5).

## C Descriptions of Baseline Explanation Methods

Graph explanation methods seek to identify a subgraph that maximally retains the predictive signal of the original graph $G$ under a budget constraint $B$ on the size of the subgraph (Ying et al., 2019). The subgraph is identified through masking out unwanted edges in the original graph, and to allow gradient-based optimization, existing works relax the discrete (hard) masks with discrete values 0 and 1 to soft masks with values between 0 and 1. Mathematically, this is expressed as:

$$G_S^* = \underset{\mathbf{M}'}{\arg\min}\ \mathcal{L}(Y; \Phi(\mathbf{X}, \mathbf{M}' \odot \mathbf{A}))$$
$$\text{s.t.} \quad \mathbf{M}' \in [0,1]^{n \times n}, \quad \sum_{i=1}^{n}\sum_{j=1}^{n} \mathbf{M}'_{ij} \leq B. \tag{C.1}$$

Practically, all the baseline explanation methods are based on this expression; the only difference is in how they realize the masks $\mathbf{M}'$.

GNNExplainer (Ying et al., 2019) directly uses a set of learnable parameters followed by a sigmoid function to represent these masks. Mathematically, $\mathbf{M}' = \sigma(\mathbf{\Theta})$, where $\sigma$ is the sigmoid function to constrain the values to be between 0 and 1 and $\mathbf{\Theta} \in \mathbb{R}^{n \times n}$ is the set of learnable parameters. It is worth noting that GNNExplainer can explain only one graph per training (the transductive setting).

PGExplainer (Luo et al., 2020) extends GNNExplainer to the inductive setting by using an MLP to generate edge masks from learned feature embeddings extracted from the final message passing layer. Once trained, PGExplainer can explain any graph from the same distribution (the inductive setting).

Learnable Randomness Injection (LRI) (Miao et al., 2022) introduces two distinct approaches: LRI-Bernoulli and LRI-Gaussian for explaining 3D geometric graphs. LRI-Bernoulli applies Bernoulli noise to nodes, that

is, applying binary masks to nodes to assess the impact of node existence on the final prediction. Similar to PGExplainer, these masks can be parametrized by MLPs. In contrast, LRI-Gaussian perturbs node positions by adding Gaussian noise, evaluating the significance of geometric features rather than identifying substructures. LRI-Gaussian is not a subgraph extraction method and is out of the scope of this work.

## D  Complexity Analysis

Compared with baselines, EDMA improves fidelity without adding additional asymptotic computational overhead. The dominant cost of generating explanations is still the forward/backward passes through the fixed 3D GNN backbone, which scale with the number of edges. Let $n$ be the number of nodes, and let $k$ denote the average number of cutoff neighbors per node (average degree), so the total number of edges satisfies $|E| \approx O(nk)$. For an optimization-based explainer that runs $S$ gradient steps, the total explanation cost is $O(S \cdot C_{\text{GNN}})$, where $C_{\text{GNN}}$ is the cost of one masked forward/backward pass. Concretely, for SchNet $C_{\text{GNN}} = O(|E|)$, while for DimeNet++ $C_{\text{GNN}} = O(|E| + D)$, where $D$ is the number of angle triplets (typically $D = O(nk^2)$). Baselines and EDMA therefore share the same leading cost term.

EDMA's additional computation is only the mask mapping from energies for each node: it computes two scalars per node to represent energies and applies a temperature-scaled sigmoid plus a stretch-and-clamp operation, all of which are elementwise operations with cost $O(n)$ (and forming edge masks from node masks costs $O(|E|)$). Since these costs are dominated by message passing term $O(|E|)$ (and $O(D)$ for DimeNet++), EDMA adds no extra asymptotic overload. The overall complexity remains $O(S \cdot |E|)$ for SchNet and $O(S \cdot (|E| + D))$ for DimeNet++.

## E  Experimental Details

To ensure a fair comparison between our proposed model and existing methods, we performed extensive hyperparameter tuning for both our approach and the baseline methods. For the baseline methods, we employed the grid search to systematically explore their respective hyperparameter spaces. In the case of the QM9 dataset, each model's performance was evaluated based on the Mean Absolute Error (MAE) on the test set, with the objective of identifying the optimal parameter configurations.

**SchNet**. For property $\mu$: we tested GNNExplainer-Dense using the following parameter settings: the coefficient for size loss varied from 1.0 to 5.5 with a step size of 2.0, the coefficient for entropy loss ranged from 0.1 to 1.1 with a step size of 0.4, and the number of training epochs ranged from 50 to 500 with a step size of 200. The optimal parameters were determined as follows: a size loss coefficient of 1.0, an entropy loss coefficient of 0.9, and 50 training epochs. Similarly, for PGExplainer-Dense, the training parameters were set with a training epoch of 40, a size loss coefficient of 30.0, and an entropy loss coefficient of 1.6. For the LRI-Bernoulli method, the training epoch was set to 50, with a prediction loss coefficient of 5.0 and an information loss coefficient of 1.0. The EDMA model's training epoch was established at 300, with the parameter $\alpha$ set to 1.0. Hyperparameters $\gamma$ and $\zeta$ are set to $-0.1$ and $1.1$, respectively, and are kept fixed for all experiments. In all experiments, we fix the initial temperature $T_0 = 5.0$ and use geometric decay with a mild exponent (0.4). We use an epoch cap to stop annealing once masks are already near-binary (to avoid over-annealing and improve stability). For SchNet on $\mu$, we use cap $= 4$ (other settings use the same schedule with a different cap).

For the property $G_f$, the following parameters were established for GNNExplainer-Dense: the coefficient for size loss was set to 300.0, the coefficient of entropy loss was also set to 300.0, and the number of training epochs was fixed at 300. In the case of PGExplainer-Dense, we set the training epoch to 100, with a coefficient of size loss of 520.0 and entropy loss coefficient of 300.0. For LRI-Bernoulli, the training epoch was established at 300, with a prediction loss coefficient of 1.0 and an information loss coefficient of 3.0. The training epoch for EDMA was similarly set to 300, with the parameter $\alpha$ assigned to a value of 500.0. Additionally, due to the presence of a shortcut embedding preceding the final readout layer in the PyG implementation for SchNet on property $G_f$, the node mask was multiplied by this embedding layer to ensure the validity of the experimental setup. For temperature, we use cap $= 25$.

**DimeNet++.** For the property $\mu$ on DimeNet++, we established the following for GNNExplainer-Dense: the coefficient for feature size loss was set to 1.5, the entropy loss coefficient was set to 0.5, and the number of training epochs was fixed at 200. In the case of PGExplainer-Dense, the training epoch was set to 150, with a size loss coefficient of 0.5 and an entropy loss coefficient of 2.5. For LRI-Bernoulli, the training epoch was set to 500, with a prediction loss coefficient of 1.0 and an information loss coefficient of 0.5. The training epoch for EDMA was similarly established at 300, with the parameter $\alpha$ assigned a value of 3.0. For temperature, we use cap $= 25$.

For the property $G_f$, we established the following parameters for GNNExplainer-Dense: the size loss coefficient was set to 0.5, the entropy loss coefficient was set to 5.0, and the number of training epochs was fixed at 300. For PGExplainer-Dense, we set the training epoch to 100, with both the size and entropy loss coefficient set to 5.0. In the case of LRI-Bernoulli, the training epoch was set to 500, with a prediction loss coefficient to 1.0 and an information loss coefficient of 5.0. The training epoch for EDMA was similarly established at 500, with the parameter $\alpha$ assigned a value of 8.0. It is important to note that while we use the same notation for $G_f$, in the PyG package for DimeNet++ this property specifically refers to the atomization free energy. For temperature, we use cap $= 35$.

**ProNet.** For the EC dataset, we established the following parameters for GNNExplainer-Dense: the size loss coefficient was set to 10.0, the entropy loss coefficient was set to 8.0, and the number of training epochs was fixed at 500. For PGExplainer-Dense, we set the training epoch to 20, with the size loss coefficient set to 0.05 and entropy loss coefficient set to 1.0. In the case of LRI-Bernoulli, the training epoch was set to 20, with both the prediction and information loss coefficient to 1.0. The training epoch for EDMA was similarly established at 500, with the parameter $\alpha$ assigned a value of 8.0. For temperature, we use cap $= 4$.

## F  Why Table 1 is Non-monotonic in $k$

Tables 2–4 exhibit the expected behavior that Fidelity$^-$ decreases as the node budget $k$ increases, whereas Table 1 does not. We investigate this discrepancy and find that the non-monotonic behavior is specific to the dipole moment prediction head used in the PyG implementation of SchNet.

For essentially all QM9 targets, increasing $k$ yields the expected trend: the prediction error between the selected substructure and the full graph decreases as more atoms are included. In our experiments, this monotonic improvement holds for all QM9 targets for DimeNet++, and for all QM9 targets for SchNet except the dipole moment $\mu$. This is consistent with the standard additive readout used by these models, where including more atoms provides a better approximation to the full-graph prediction (see Tables 2–4).

In contrast, SchNet's PyG implementation for dipole moment $\mu$ uses a geometry-dependent readout: after producing per-atom features, it computes the molecule's center of mass and weights each atom's contribution by its relative coordinate to this center before aggregating to a scalar. Under our evaluation protocol, we feed only the selected top-$k$ atoms into the backbone to compute the substructure prediction. As $k$ varies, the center of mass of the selected subgraph (and thus the geometry-dependent weights) can shift non-monotonically. Therefore, unlike the additive readout case where the MAE primarily depends on the selected atoms, here the MAE also depends on how the subgraph's center of mass deviates from the full molecule's center of mass, and there is no analytical guarantee that Fidelity$^-$ decreases monotonically with $k$, as shown in Table 1.

Importantly, this phenomenon affects all explainers under this evaluation protocol, since it is induced by the backbone implementation. Therefore, the meaningful comparison in Table 1 is relative across explainers under the same backbone and evaluation setting. Under this shared setting, EDMA still consistently improves Fidelity$^-$ compared to other baselines across budgets.

## G  Comparison with GEM

To further validate the effectiveness of EDMA in mitigating the soft-to-discrete discrepancy, we compare our method against GEM (Lin et al., 2021). Unlike soft mask methods, GEM is a causal explanation approach (motivated by Granger causality) that incrementally selects edges to form an explanatory sub-

graph. Importantly, GEM performs discrete selection rather than optimizing soft masks, and thus avoids the relaxation-induced soft-to-discrete discrepancy. Since GEM is originally formulated for the edge selection, we adapt it to the node selection to ensure a fair comparison. As shown in Table 9, EDMA consistently outperforms GEM across most node budgets ($k$) on the SchNet backbone for the property $\mu$, which further demonstrates the faithful explanation predicted by EDMA.

Table 9: Explanation Fidelity$^-$ (the lower the better) for all baseline methods, GEM, and our proposed EDMA method using SchNet as the backbone on the property $\mu$ (dipole moment, in Debye (D)). The best results are highlighted in bold.

| Top-$k$ | 2 | 3 | 4 | 5 | 6 | 7 | 8 | 9 |
|---|---|---|---|---|---|---|---|---|
| GNNExplainer-Dense | 3.88 | 5.62 | 7.28 | 8.05 | 8.27 | 8.00 | 7.59 | 6.87 |
| PGExplainer-Dense | 2.91 | 3.73 | 4.83 | 6.09 | 6.62 | 6.55 | 6.81 | 6.08 |
| LRI-Bernoulli | 3.50 | 4.84 | 6.16 | 6.88 | 7.10 | 7.29 | 7.43 | 7.32 |
| GEM | 2.76 | **3.66** | 4.55 | 5.13 | 5.69 | 6.26 | 6.31 | 5.89 |
| **EDMA** | **2.74** | 3.73 | **4.31** | **4.83** | **5.08** | **5.47** | **5.72** | **5.31** |

## H  Additional Qualitative Results

We provide additional qualitative results and chemical explanations for the dipole moment. For each molecule, candidate functional groups are extracted from the molecular structure, and domain experts identify the functional groups relevant to the dipole moment, together with brief chemical justifications. These annotations are generated independently of all explanation methods and are used solely as qualitative references for comparison. The node budget of each explanation method is matched to the number of atoms in the corresponding annotated functional groups. Following the setting in the main paper, explanatory subgraphs are not required to be connected, since chemically relevant functional groups may consist of multiple spatially separated regions. The selected atoms are visualized on the original QM9 molecular structures. Fig. 5 and Fig. 6 present additional visualization results, while Fig. 7 provides the corresponding chemistry-based descriptions. Consistent with the observations in Sec. 4.3, EDMA more frequently identifies explanatory substructures that align with chemically meaningful functional groups.

## I  Invariant and Equivariant GNNs

When modeling molecular representations, two main approaches can be used: invariant GNN models and equivariant models. The key difference between these two categories lies in how they handle geometry transformations, such as rotation and translation. Invariant models aim to capture atomic representations that are invariant under these geometric transformations. Typically, these invariant representations are achieved by incorporating relative 3D information as scalar inputs, such as Euclidean distances between atom pairs, bond angles, and torsion angles between planes. During the message passing process, these learned invariant representations are used to exchange messages between atoms. In this manner, the final output remains invariant under geometric transformations. SchNet and DimeNet++ are classical examples of invariant models: SchNet (Schütt et al., 2017) incorporates pairwise distances in its representations, while DimeNet++ (Gasteiger et al., 2020b;a) includes additional bond angles to improve expressiveness.

In contrast, equivariant models seek to represent features that are equivariant to the geometry transformations of the input. When the input graph undergoes geometric transformations, the learned representations transform in a predictable manner. High-order tensors, such as spherical harmonics, are often used to represent these features to ensure equivariance. In the message passing process, tensor products are commonly applied for tensor operations, while Clebsch-Gordan coefficients and Wigner-D matrices are frequently used to enforce equivariance. In our study, the classical equivariant model EGNN (Satorras et al., 2021) serves as a baseline for comparing the performance of our method and GNNExplainer. Extensive experimental results demonstrate that, whether using invariant or equivariant models, our model achieves competitive performance.

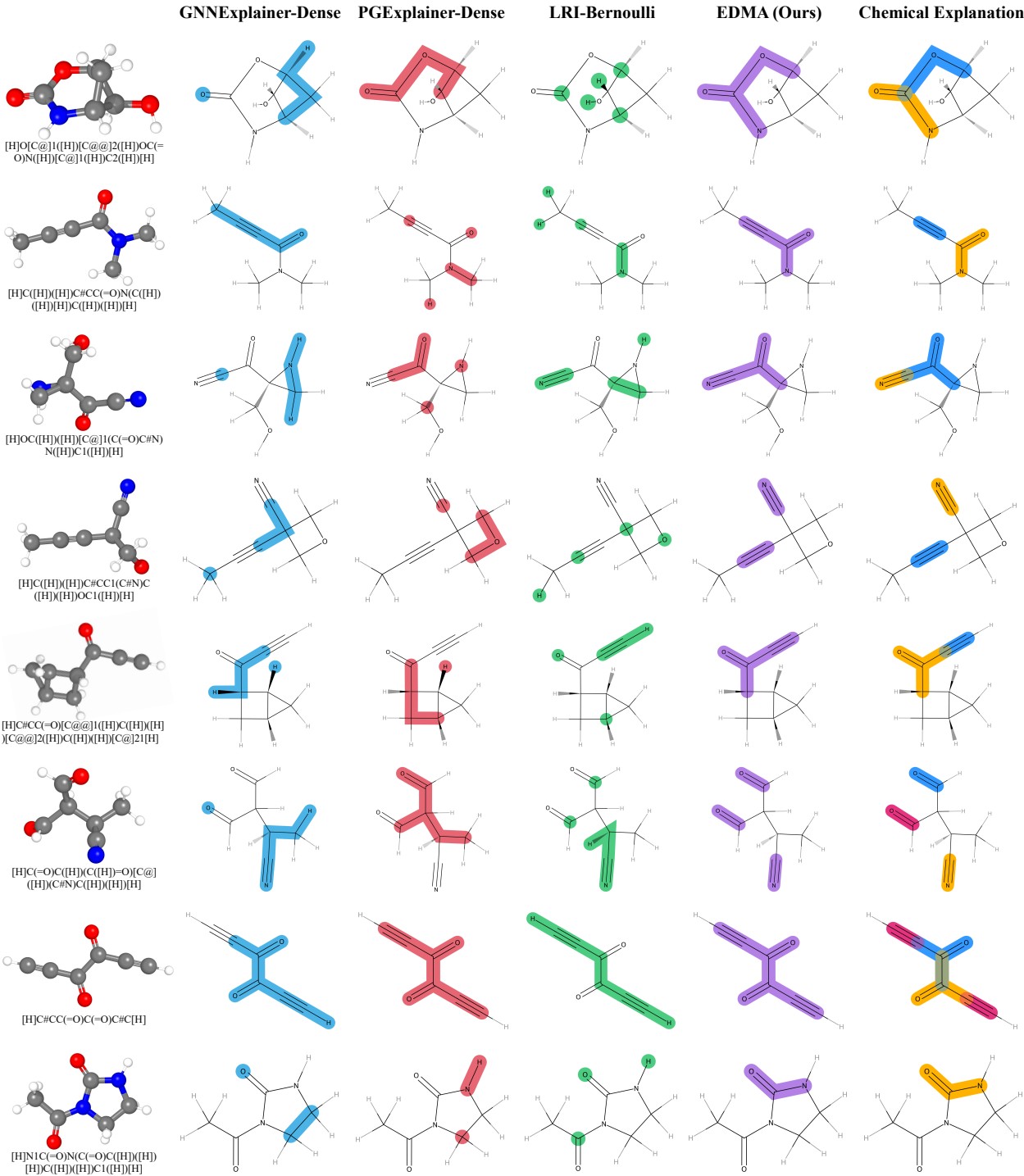

Figure 5: Additional qualitative results on molecules from the QM9 for the dipole moment. The first column showcases 3D molecules from the QM9 dataset, along with corresponding SMILES strings. The following columns present the explanation results from various baseline methods alongside our EDMA method. Finally, provides chemical interpretations, where distinct colors are used to differentiate the contributing functional groups.

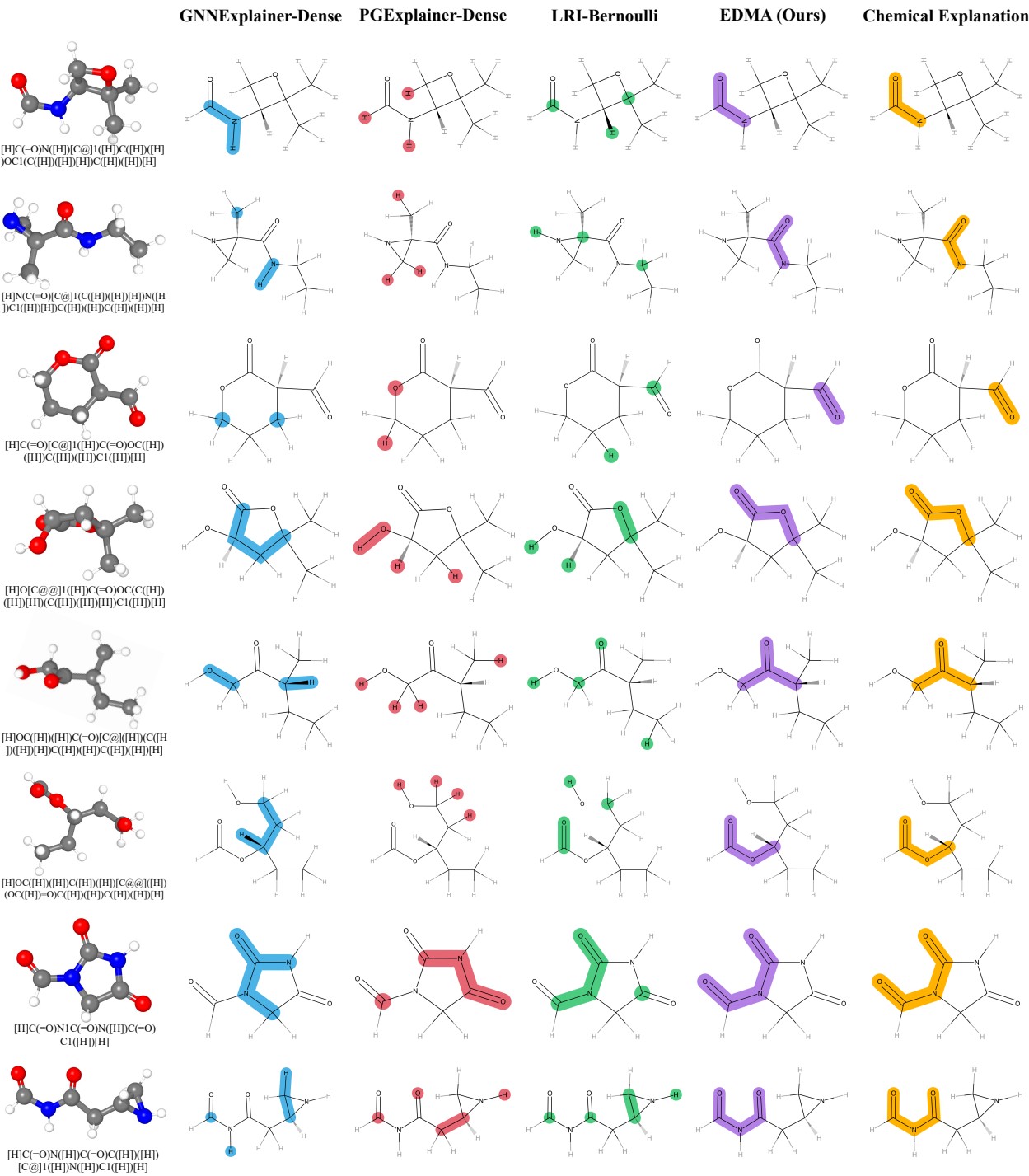

Figure 6: Additional qualitative results on molecules from the QM9 for the dipole moment. The first column showcases 3D molecules from the QM9 dataset, along with corresponding SMILES strings. The following columns present the explanation results from various baseline methods alongside our EDMA method. Finally, the last column provides chemical interpretations, where distinct colors are used to differentiate the contributing functional groups.

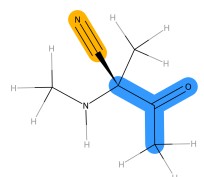

The nitrile group provides a strong dipole component through the C≡N bond polarity. The ketone carbonyl adds a further vector at an angle to the nitrile axis, enhancing the net dipole moment.

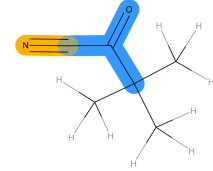

The nitrile group represents a key source of polarity. Together with the adjacent ketone, these groups notably shape the net dipole due to their similar orientations.

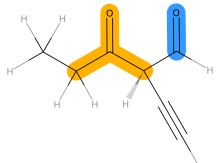

The ketone and aldehyde carbonyls are important polar groups. Their constructively aligned bond dipoles significantly shape the net molecular dipole moment.

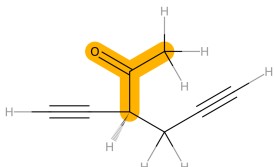

The ketone carbonyl creates a significant dipole component through its localized and highly polar C=O bond.

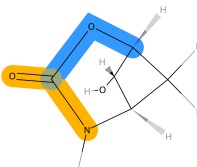

The lactam-like unit is an important contributor, where the nitrogen and carbonyl group act as a single polar entity. The ring oxygen provides further contributions via its local bond dipoles.

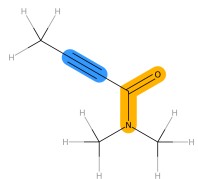

The amide group serves as a key polar moiety due to resonance-induced polarization. The alkyne adds a subtle directional component to the total dipole moment.

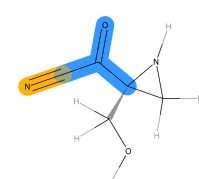

The nitrile group serves as a key polar moiety, bonded to the ring-attached carbonyl. These groups provide significant dipole components that collectively shape the net molecular moment.

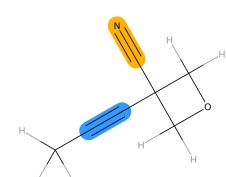

The nitrile group represents a significant polar moiety. The internal alkyne provides a notable contribution arising from its asymmetric chemical environment.

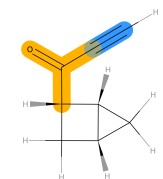

The ketone carbonyl represents a significant polar moiety. The terminal alkyne provides a notable dipole component, with its orientation determined by the rigid bicyclic framework.

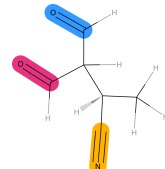

The nitrile group acts as a significant polar moiety. Spatially separated aldehyde carbonyls (blue, raspberry) provide notable dipole components that reinforce the net molecular moment.

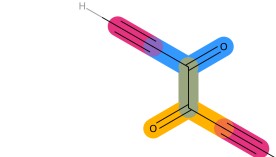

The ketone carbonyls (amber, blue) represent significant polar moieties. The terminal alkynes provide notable dipole components that influence the net moment.

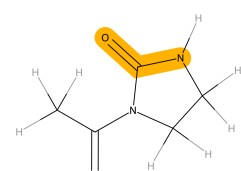

The amide group represents a significant polar moiety. Its resonance-driven dipole provides a notable directional component to the molecular dipole moment.

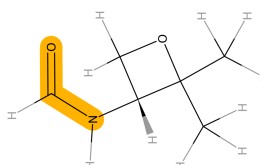

The amide group represents a significant polar moiety. Its resonance-driven dipole provides a notable directional component to the molecular dipole moment.

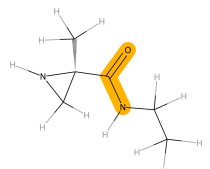

The amide moiety is a significant polar contributor, where its strong bond dipole defines a notable component of the total moment.

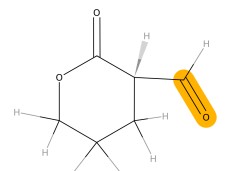

The aldehyde carbonyl represents a significant polar moiety, characterized by its strong and highly directional C=O bond dipole.

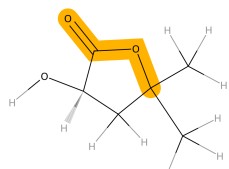

The lactone moiety represents a significant polar source. Its integrated carbonyl and ring oxygen dipoles provide a notable component to the net molecular dipole.

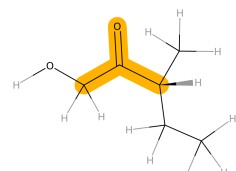

The ketone carbonyl represents a significant polar moiety. Its localized bond dipole provides a notable component to the net molecular dipole moment.

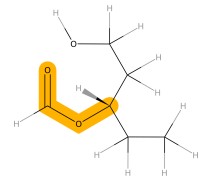

The formate ester moiety represents a significant polar source. Its dipole contribution is driven by the highly polarized bonds within the formate functional unit.

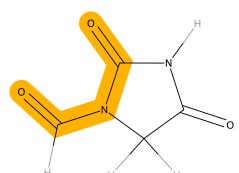

The carbonyl-nitrogen moiety creates significant polarity through resonance-induced polarization, providing a notable component to the dipole moment.

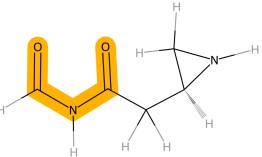

Electronic resonance within the carbonyl-nitrogen moiety creates a significant dipole, serving as a notable component of the total molecular moment.

Figure 7: Functional-group-level chemical annotations and corresponding chemistry-based explanations for the molecules in visualization. Different colors indicate distinct functional groups that are considered relevant to the molecular dipole moment. These annotations serve as qualitative references for interpretation.

