# OpenReview forum: "Confidence-Aware Explanations for 3D Molecular Graphs via Energy-Based Masking"
_TMLR — Accepted by TMLR_

### Review · Reviewer_2nsV · 2026-02-05

**Summary Of Contributions:**

This paper identifies a fundamental bottleneck in existing 3D GNN explanation methods—the mismatch between optimized soft masks and final discrete explanatory subgraphs, which is amplified in dense 3D molecular graphs. To address this issue, the authors propose EDMA, an energy-based framework that enforces mask discreteness by design via energy separation, leading to more confident and faithful explanations. Experiments on molecular and protein datasets with multiple 3D GNN backbones show consistent improvements in explanation fidelity.

### Strengths
- **Clear and well-motivated problem formulation.**
  The paper clearly identifies a fundamental bottleneck in 3D GNN explanations—the gap between optimized soft masks and final discrete explanatory subgraphs in dense 3D graphs—and proposes an energy-based explanation function to address this issue.

- **Reasonable method design.**
  The upper-bound decomposition in Eq.(5), separating the soft-mask loss and the soft-to-discrete discrepancy, provides a clear explanation of why existing methods may optimize well yet produce poor explanations.

- **Sufficient experimental validation.**
  The method is evaluated on both QM9 and EC datasets using multiple 3D GNN backbones, and consistently achieves improved explanation fidelity. The results are further supported by ablation studies and qualitative analyses.

### Weaknesses
- **Limited theoretical formalization.**
  The error bound in Eq.(5) is mainly a conceptual decomposition rather than a strict theoretical guarantee. A more formal analysis would strengthen the theoretical contribution.

- **Lack of comparison with alternative discretization mechanisms.**
  While the experiments show that mask discreteness is critical, the energy-based formulation is not the only possible approach. Comparisons with other discretization techniques (e.g., hard-concrete, ℓ₀ regularization, or Gumbel-Softmax) would provide additional insight.

- **Missing computational complexity analysis.**
  The paper does not explicitly discuss the computational overhead or efficiency of the proposed method compared to existing explainers.

**Audience:**

Yes

**Audience Explanation:**

The findings of this paper can facilitate drug editing and molecular optimization tasks, thereby enabling more efficient drug discovery, and may also be beneficial for explanation analysis in other 3D graph domains.

**Broader Impact Concerns:**

None.

**Claims And Evidence:**

Yes

**Claims Explanation:**

Overall, the paper’s main claims and method design are supported by extensive experimental results.

**Requested Changes:**

- **Strengthen theoretical analysis:** Clarify assumptions and provide additional insights (e.g., energy gap vs. mask entropy/variance or a toy theoretical example).
- **Compare alternative discretization methods:** Include baselines such as hard-concrete or Gumbel-Softmax under the same budget and backbone.
- **Add efficiency and stability analysis:** Report explainer runtime, memory overhead, and sensitivity to the temperature parameter \(T\).

---

> ### Author Response · Authors · 2026-03-20
> **Response to Reviewer 2nsV - Part I**
>
> Thank you very much for your valuable feedback! We have revised the manuscript and provide our detailed responses below.
>
> - **W1: Limited theoretical formalization**
>   - **We address this concern by expanding the analysis of Eq. (5) and incorporating these revisions into Appendix A**. Specifically, Appendix A.1 provides a detailed derivation of Eq. (5) under the MAE loss, and Appendix A.2 proves Lipschitz continuity with respect to the edge masks. Building on these results, Appendix A.3 derives an explicit discrepancy bound that further supports Eq. (5). We summarize the key results of Appendix A.2 and Appendix A.3 as follows.
>   - **Lipschitz continuity w.r.t. the edge mask**
>     - In Appendix A.2, we prove that a broad class of 3D GNNs is Lipschitz continuous with respect to the edge mask. Concretely, for typical 3D GNNs that construct edges via a cutoff radius, there exists a constant $L_\Phi<\infty$ such that for any two soft edge masks $\tilde{\mathbf M}$, $\tilde{\mathbf M}'\in[0,1]^{n\times n}$,
>     $
>     \big|\hat{Y}(\tilde{\mathbf M})-\hat{Y}(\tilde{\mathbf M}')\big|
>     \le
>     L_\Phi \|(\tilde{\mathbf M}-\tilde{\mathbf M}')\odot \mathbf{A}\|_{1,E}.
>     $
>     - Here, $\mathbf A$ is the adjacency matrix, $\odot$ is the elementwise (Hadamard) product, $\lVert\cdot\rVert_{1,E}$ is the edge-wise $\ell_1$ norm, and $\hat Y(\tilde{\mathbf M})$ denotes the 3D GNN prediction obtained by masking message passing using $\tilde{\mathbf M}$. The constant $L_\Phi$ is finite and depends on the layerwise Lipschitz constants as well as graph density (via the maximum node degree $d_{\max}$). We further discuss how SchNet and DimeNet(++) fit into this analysis.
>
>   - **Mask amplification effect in 3D graphs**
>     - In Appendix A.3, for a graph with $n$ nodes, we use the node-mask parameterization $\mathbf M'=\boldsymbol m\otimes \boldsymbol m$ with node masks $\boldsymbol m\in[0,1]^n$, and its discretization $\mathbf M=\boldsymbol m^d\otimes \boldsymbol m^d$ with $\boldsymbol m^d \in \lbrace 0,1\rbrace^n$. $\otimes$ denotes the outer product. We derive
>     $
>     \lVert(\mathbf M'-\mathbf M)\odot \mathbf A\rVert_{1,E}
>     \le
>     2\sum_{i=1}^n \deg(i)|m_i-m_i^d|
>     \le
>    2d_{\max}\|\boldsymbol m-\boldsymbol m^d\|_1 .
>    $
>
>     - Here, $\deg(i)$ is the degree of node $i$ in the induced graph, and $d_{\max}$ denotes the maximum node degree. The term $|m_i-m_i^d|$ measures the discrepancy between the soft mask and the corresponding discrete one. This bound reveals a degree amplification effect: a small per-node discrepancy $|m_i-m_i^d|$ is multiplied by $\deg(i)$, and the total discrepancy further accumulates over all nodes.
>     - In 3D graphs, the model performance usually improves with a larger cutoff, which increases graph density and node degrees. As a result, even modest per-node discrepancies can lead to a large overall soft-to-discrete discrepancy in dense 3D graphs.
>
>   - **Explicit quantitative upper bound for Eq. (5)**
>     - Combining the above formulations yields an explicit quantitative upper bound on the discrepancy term in Eq. (5):
>     $
>     \big|\hat Y(\mathbf M')-\hat Y(\mathbf M)\big|
>    \le
>    2L_\Phi d_{\max}\|\boldsymbol m-\boldsymbol m^d\|_1.
>    $
>     - This provides a concrete upper bound and formalizes why high-fidelity explanation is more challenging in dense 3D graphs: both $L_\Phi$ and $d_{\max}$ can grow with the cutoff radius, amplifying small non-binary mask values.
>
>   - **Applicability to practical 3D GNNs**
>     - Under mild architectural assumptions (finite cutoff neighborhoods, bounded geometric features, and Lipschitz message/update/readout functions), our analysis applies to a broad class of 3D molecular GNNs. We also discuss why these assumptions are reasonable for common 3D GNNs and use SchNet and DimeNet(++) as concrete examples. Therefore, the derived bounds are directly relevant to widely used 3D GNN architectures.
>   - Overall, Appendix A strengthens Eq. (5) by turning the decomposition into an explicit quantitative bound: **(i) a Lipschitz control of the prediction with respect to the edge mask, and (ii) a degree amplification effect showing how edge mask discrepancy grows with the gap between soft and discrete node masks.** This also explains why dense 3D graphs are particularly challenging to explain when high fidelity explanations are desired.

---

> ### Author Response · Authors · 2026-03-20
> **Response to Reviewer 2nsV - Part II**
>
> - **W2: Lack of comparison with alternative discretization mechanisms**
>   - In the manuscript, Sec. 3.4 discusses limitations of regularization-based discretization, and Table 7 provides experimental results on property $\mu$ with SchNet. Specifically, we remove the entropy regularization from the baseline loss functions (keeping only the $\ell_1$-style budget term) and denote these variants with “\*”.
>     - Results in Table 7 show that removing the entropy regularization does not lead to noticeable performance degradation for baselines, and both the original and “\*” variants remain less competitive than EDMA. This suggests that the $\ell_1$ regularization alone is not sufficient to reliably resolve the issue for dense 3D graphs.
>   - **To further address the concern more directly, we add Appendix B** to provide a theoretical comparison between EDMA and alternative discretization relaxations (e.g. $\ell_0$-style regularization, hard-concrete, and Gumbel-Softmax/Concrete).
>     - The key point is that EDMA provides explicit control of the soft-to-discrete distance through the energy-based parameterization, which directly tightens the discrepancy term amplified by dense graphs.
>     - By contrast, **$\ell_0/\ell_1$-style penalties mainly promote sparsity**, but do not directly control the remaining masks' soft-to-discrete distance (so intermediate values can persist), while **hard-concrete and Gumbel-Softmax** rely on stochastic gates and are typically justified at expectation level, without the same per-instance guarantee unless additional tuning is performed.
>     - In this sense, **EDMA deterministically targets the quantity amplified by dense 3D graphs**, complementing the empirical results in Table 7.
>
> - **W3: Missing computational complexity analysis**
>   - **We address this concern by providing the complexity analysis in Appendix C**, and we summarize it below.
>   - Compared with baselines, **EDMA improves fidelity without adding additional asymptotic computational overhead.** The dominant cost of generating explanations is the forward/backward passes through the fixed 3D GNN backbone, which scale with the number of edges.
>     - Let $n$ be the number of nodes and let $k$ denote the average number of cutoff neighbors per node (average degree), so the total number of edges satisfies $|E|\approx O(nk)$. For an optimization-based explainer that runs $S$ gradient steps, the total explanation cost is $O(S\cdot C_{\text{GNN}})$, where $C_{\text{GNN}}$ is the cost of one masked forward/backward pass.
>     - Concretely, for SchNet $C_{\text{GNN}} = O(|E|)$, while for DimeNet++ $C_{\text{GNN}} = O(|E| + D)$, where $D$ is the number of angle triplets (typically $D= O(nk^2)$). Baselines and EDMA therefore share the same leading cost term.
>     - EDMA's additional computation is only the mask mapping from energies for each node: it computes two scalars per node to represent energies and applies a temperature-scaled sigmoid plus a stretch-and-clamp operation, all of which are elementwise operations with cost $O(n)$ (and forming edge masks from node masks costs $O(|E|)$). Since these costs are dominated by message passing term $O(|E|)$ (and $O(D)$ for DimeNet++), EDMA adds no extra asymptotic overhead. The overall complexity remains $O(S\cdot |E|)$ for SchNet and $O(S\cdot(|E| + D))$ for DimeNet++.

---

> > ### Comment · Reviewer_2nsV · 2026-04-04
> > **My concern has been addressed.**
> >
> > Thanks for the author's response. My concern has been addressed.

---

### Review · Reviewer_jjhE · 2026-02-16

**Summary Of Contributions:**

This work aims to explain predictions in graph neural networks (GNNs) via extraction of relevant subgraphs. The authors propose an explanation strategy for GNNs on 3D molecular graphs with distance based cutoff (mostly for energy, force, molecular property prediction). The particular setting of a GNN predicting energies and forces is a special case, where predictions are governed by physical laws. This inspires the design of an "energy"-based interpretability technique. Common interpretability strategies used for 2D graphs (graphs containing molecular bonds as edges) assume sparse edges and therefore cannot be transferred successfully to locally dense 3D graphs (more edges due to graph creating as cutoff-graph). The authors propose building an explainer network to predict two "energy values" for each edge: a larger and a smaller one. Depending on a temperature hyperparameter and the predicted energies, a probability of the edge ending up in the explanation subgraph is computed. A loss function consisting of two opposing terms is used to limit the number of edges in the predicted subgraph while retaining high accuracy for the task at hand.

Strengths:
- The problem with transferring interpretability techniques from bond-graphs to more densely connected 3D graphs is well motivated and addressed
- As far as I can conclude, the authors are the first to address the rather challenging problem of generating explanations for 3D molecular graphs (there might be relevant work which I am unaware of)
- The method performs well in empirical studies, however being only compared to methods designed for 2D bond-graphs

Weaknesses:
- There are no units given for experimental results and no absolute baseline performance is being reported, which makes it difficult to judge the experimental results, especially since it is only possible to benchmark against 2D interpretability methods, which are not designed for this task
- Certain design choices, especially the choice to predict two "energy values" instead of one lack thorough justification and ablation
- The example in Figure 4 illustrates identified contributions of bonds to a global vector observable: the Dipole Moment. I find the sparse mask inherently ambiguous here, since the dipole moment as vectorial quantity depends not only on local charges, but also on the physical separation of those charges, i.e. the scaffold to hold them in place. Therefore, the whole molecule should influence the dipole moment to a large extent. Maybe it would be better to repeat this experiment and predict the partial charges instead (should also be in the dataset).

**Audience:**

Yes

**Audience Explanation:**

I imagine, that the presented explanation strategy could be useful for practitioners to build better Machine Learning Force Fields (MLFFs) being able to debug their models using explanations to identify possibly unphysical behavior. The authors mention possible applications to drug discovery or related fields, but there are no concrete statements to be found, how exactly this application could look like. In general, I feel the authors could include more qualitative studies to showcase how their method helps to understand the impact of the molecular geometry on some properties in the QM9 and EC datasets. For reasons stated above, I think that the dipole moment is not very well suited for this analysis. In general, I find the direction of identifying the impact of certain molecular building blocks on properties of the whole structure very interesting.

**Broader Impact Concerns:**

I have no concerns about any ethical implications that would necessitate a separate Broader Impact statement.

**Claims And Evidence:**

Yes

**Claims Explanation:**

The theory of moving from 2D bond graphs to 3D point clouds (with local edge connectivity via cutoffs), is well justified and supported by empirical evidence. The authors perform experiments on 2 datasets using 3 different reference architectures (ProNet, DimeNet++, Schnet), using 3 different prediction targets. While this seems exhaustive, I must strongly emphasize the following points that should definitely be improved:
- The tables report relative improvements or unit-less error metrics. For all quantities, where units are available those must be included, and as well absolute error numbers (MAE) for the performance of the baseline models. Otherwise it is really difficult to judge the efficacy of the method.
- There is a counter intuitive trend comparing Tables 1 and 3: only by changing the underlying model architecture (Schnet > DimeNet++), the explanation technique works much worse for increasing k. In fact the trend is exactly opposite (error increasing for Schnet, while decreasing for DimeNet++). This needs further investigation and explanation in the text.
- The choice to predict two "energy values" instead of one is unjustified, since only the difference of both enters in eq. 6. This should be clearly motivated and ablated.
- For completeness, the authors should add a version of their method with entropy regularization in table 7

**Requested Changes:**

**Must have (decreasing priority):**
- Improve tables with absolute baseline results and units
- Explain and investigate trends for increasing k
- Ablate the impact of predicting two "energy values" instead of the difference directly
- Add a version of EDMA with entropy regularization in table 7

**More suggestions or questions:**
- Since this paper investigates small molecules, it can be assumed that most of the atoms fall within the cutoff range of each other, which supports the claim that graphs are complete or "dense". However, for larger molecules this is not the case, and the adjacency matrix becomes only locally dense. It would be interesting to evaluate the method in this setting, or at least hypothesize if it still would work.
- Why is it that in fig 4 we have different numbers of edges for the different methods? Is it not top k edges for all methods that are kept?
- How did you tune the temperature hyperparameter T for your method?
- Especially for readers, who might be unfamiliar with the field, it would be nice to include a short description of how the two "energy values" are predicted - I assume a separate network is trained, which is optimized using the loss in equation (7)
- Section 3.4 introduces some new details about related work, which are missing in section 2 - I think it would be easier to follow for the reader if entropy regularization was introduced before already

---

> ### Author Response · Authors · 2026-03-20
> **Response to Reviewer jjhE - Part I**
>
> Thanks a lot for your valuable comments! We have revised the manuscript and provide point-by-point responses below.
> - **Q1: Improve tables with absolute baseline results and units.**
>   - Thanks for pointing this out. **In the revision, we add explicit units for the tables.** We also report the absolute test MAE of the underlying pretrained backbone models below to provide an absolute performance reference.
>   - For dipole moment $\mu$, the unit is Debye (D). For free energy at 298.15K $G_f$, the unit is eV. We use the pretrained weights and evaluation settings provided by the PyG package. On the QM9 test split, the full-graph test MAEs of the backbone models are:
>     - SchNet: 0.033 D for dipole moment $\mu$; 0.012 eV for free energy $G_f$.
>     - DimeNet++: 0.030 D for dipole moment $\mu$; 0.007 eV for atomization free energy $G_f$.
>   - This provides an absolute baseline for interpreting the explanation fidelity results reported in Tables 1–4. Note that fidelity values in Tables 1–4 are errors of substructure-based predictions (Top-$K$ nodes), and are therefore expected to be larger than the full-graph.
> - **Q2: Explain and investigate trends for increasing $K$.**
>
>   - We thank the reviewer for pointing out the seemingly counterintuitive trend between Table 1 and Table 3. We investigated this and found that **the non-monotonic behavior as the budget $K$ increases is specific to the dipole moment prediction head used in the PyG implementation of SchNet.**
>   - For all QM9 targets we consider in SchNet and DimeNet++, except $\mu$ in SchNet, the models use an additive readout, i.e., the prediction is obtained by aggregating per-atom contributions. Under this setting, increasing the node budget $K$ is expected to reduce the MAE and hence improve $\mathrm{Fidelity}^{-}$ monotonically, as reflected in Tables 2–4.
>   - In contrast, SchNet’s PyG implementation for dipole moment $\mu$ uses a geometry-dependent readout: after producing per-atom features, it computes the molecule’s center of mass and weights each atom’s contribution by its relative coordinate to this center before aggregating to a scalar.
>     - Under our evaluation protocol, we feed only the selected top-$K$ atoms into the backbone to compute the substructure prediction. As $K$ varies, the center of mass of the selected subgraph (and thus the geometry-dependent weights) can shift non-monotonically.
>     - Therefore, **unlike the additive readout case where the MAE primarily depends on the selected atoms, here the MAE also depends on how the subgraph’s center of mass deviates from the full molecule’s center of mass**, and there is no analytical guarantee that $\mathrm{Fidelity}^{-}$ decreases monotonically with $K$.
>     - Importantly, this effect is induced by the backbone model and the subgraph-evaluation protocol, so it affects all explainers similarly under the same setting. Therefore, the meaningful comparison in Table 1 is the relative performance across explainers under identical backbone and evaluation conditions, where EDMA still consistently improves $\mathrm{Fidelity}^{-}$ compared to other baselines. **We have added this clarification to the revised manuscript in Appendix E.**

---

> > ### Author Response · Authors · 2026-03-20
> > **Response to Reviewer jjhE - Part II**
> >
> > - **Q3: Ablate the impact of predicting two "energy values" instead of the difference directly.**
> >   - The key reason we use two energies is that it provides a well-defined two-state model and therefore a principled probability mapping.
> >     - **With two energies $\mathcal{E}_0(\mathbf e_i)$ and $\mathcal{E}_1(\mathbf e_i)$, the mask probability in Eq. (6) follows directly from a two-state normalization.**
> >     - In contrast, if we use only a single value per node, there is no unique way to map it to a probability without introducing an additional reference level. For example, using a softmax implicitly fixes the reference energy to $0$; using a sigmoid requires an explicit threshold $b$: $m_i(T)=\sigma((e_i-b)/T)$. In both cases, the probability is no longer determined purely by the learned energy, but also by the chosen reference, which can affect optimization across different tasks and backbones.
> >     - **Using two energies removes this ambiguity because only the energy gap $\mathcal{E}_1(\mathbf e_i)-\mathcal{E}_0(\mathbf e_i)$ matters, and any common shift cancels out.** Therefore, the mapping in Eq. (6) is invariant to energy offsets and directly supports the “confidence via energy gap” interpretation.
> >   - We ablate this design by implementing EDMA-Single (one scalar per node with an implicit reference $0$) in Table 1. We observe a consistent performance drop relative to EDMA across budgets, while EDMA-Single still improves over the baselines. This confirms that the two-energy parameterization resolves the probability-mapping ambiguity and yields more robust optimization behavior and better explanation fidelity.
> >
> >     Table 1: Explanation $\mbox{Fidelity}^{-}$ (the lower the better) on the property $\mu$ (dipole moment, in Debye (D)) with SchNet as the fixed 3D GNN backbone being explained, comparing baselines, EDMA-Single, and EDMA. For each $K$, the best results are boldfaced, and the second-best results are marked with $^{\dagger}$.
> >     |K|2|3|4|5|6|7|8|9|
> >     |---:|---:|---:|---:|---:|---:|---:|---:|---:|
> >     |GNNExplainer-Dense| 3.88 | 5.62 | 7.28 | 8.05 | 8.27 | 8.00 | 7.59 | 6.87 |
> >     |PGExplainer-Dense | 2.91$^{\dagger}$ | **3.73**| 4.83 | 6.09 | 6.62 | 6.55 | 6.81 | 6.08|
> >     |LRI-Bernoulli | 3.50 | 4.84 | 6.16 | 6.88 | 7.10 | 7.29 | 7.43 | 7.32 |
> >     |EDMA-Single| 3.22 | 3.77 |4.74$^{\dagger}$| 5.38$^{\dagger}$ | 5.65$^{\dagger}$| 5.83$^{\dagger}$ |5.91$^{\dagger}$ | 5.90$^{\dagger}$ |
> >     |**EDMA** | **2.74** | **3.73** |**4.31**| **4.83**| **5.08** | **5.47** |**5.72**| **5.31**|
> >
> > - **Q4: Add a version of EDMA with entropy regularization in Table 7.**
> >   - We thank the reviewer for this suggestion. We argue that adding an explicit entropy regularizer in Table 7 is unnecessary.
> >     - Table 7 is designed to answer our second ablation question: is there a clear need for methods that achieve discreteness by design, such as our proposed EDMA, over regularization-based approaches? To this end, we remove the entropy regularization term from the baseline methods, and we mark the resulting variants with $^{*}$. **For clarity, EDMA in Table 7 uses our original objective (Eq. (7)) and does not include any additional entropy regularizer.** As shown in Table 7, removing the entropy term does not lead to a noticeable decline in performance for baselines. This suggests that entropy regularization alone is not sufficient to achieve the desired discreteness and fidelity.
> >     - Second, Appendix B provides theoretical support showing that **EDMA’s formulation admits an entropy-regularized variational interpretation** and yields explicit control of the soft-to-discrete distance. Hence, EDMA achieves discreteness by design without requiring an additional regularizer, and adding one explicitly would be redundant.
> > - **Q5: Evaluate or explain whether the model works on large molecules.**
> >   - We agree that for larger molecules, cutoff-based graphs are typically only locally dense. However, as the number of atoms increases, the number of edges can still become very large. Our analysis in Appendix A shows that when many edges are induced, the discrepancy can be amplified and accumulated, so explanations for larger molecules can face similar challenges.
> >   - To evaluate this setting, we additionally test EDMA on the EC dataset (Sec. 4.2), which contains substantially larger and more complex molecular graphs than QM9. **With a cutoff $r_c = 10\text{Å}$, a graph with 1150 nodes contains $\sim20,000$ edges.** This is exactly the regime where optimization-based soft-mask baselines can become less faithful due to the density amplification characterized in Appendix A.
> >   - Consistent with our analysis, **Table 5 shows that only our method can faithfully explain over half of the graphs in the dataset**, which illustrates the ability of EDMA to generalize to larger and more complex structures. We will study whether our method can scale to even larger and more complex structures in future work.

---

> > > ### Author Response · Authors · 2026-03-20
> > > **Response to Reviewer jjhE - Part III**
> > >
> > > - **Q6: Different numbers of edges for different explanation methods in Fig 4.**
> > >   - We would like to clarify the misunderstanding here. Our work focuses on explaining 3D molecular models, and **we define an explanatory substructure as a subset of nodes (atoms), rather than a subset of edges.** This design choice is motivated by the reasons discussed in Sec. 3.1.
> > >     -   3D molecular systems are physically structured and governed by force field theory [1,2], and molecular properties (especially energies) can be interpreted through atomic partitioning. To faithfully explain the model’s behavior, as stated in Sec. 3.2, we place masks on nodes and select nodes accordingly as the explanatory subgraphs.
> > >     - In Fig. 4, we enforce the same budget $K$ in terms of nodes for all methods. **The number of edges shown can differ because edges are induced by the selected nodes when visualizing the substructure using the chemical bonds in QM9.** As stated in Sec. 4.3, for each molecule, we match the number of atoms identified by the explanation methods to those in the chemical explanation. This ensures a fair comparison across methods.
> > > - **Q7: The choice of hyperparameter $T$.**
> > >   - We use a temperature annealing schedule to control mask discreteness during optimization. A higher temperature produces smoother masks early on (more stable gradients and easier optimization), while decreasing the temperature later drives masks toward $\{0,1\}$, directly reducing the soft-to-discrete discrepancy.
> > >   - In all experiments, we fix the initial temperature to 5.0 and use geometric decay with a mild exponent (0.4) so the temperature decreases smoothly. We then apply an epoch cap to stop annealing once masks are already near-binary, which avoids over-annealing and improves stability. We select the cap epoch via a small validation-based search, and use: $\text{cap}=4$ for $\mu$ with SchNet, $\text{cap}=25$ for $G_f$ with SchNet, $\text{cap}=25$ for $\mu$ with DimeNet++, and $\text{cap}=36$ for $G_f$ with DimeNet++.
> > >   - **We provide the detailed settings in Appendix D.**
> > > - **Q8: Description of how the two "energy values" are predicted.**
> > >   - In EDMA, the two energy values are implemented as **learnable per-node parameters optimized per input-graph instance** (rather than being produced by an additional neural network).
> > >     - These parameters are optimized by minimizing the objective in Eq. (7) while keeping the backbone models frozen. The learned energies are converted to soft node masks via the temperature-scaled sigmoid, and the corresponding edge masks are obtained through the outer-product. This parameterization keeps the energy module lightweight and avoids introducing extra model capacity.
> > >     - Alternatively, the same formulation can be implemented with a small neural network without changing the EDMA mechanism.
> > > - **Q9: Details about related work in Sec. 3.4 are missing in Sec. 2.**
> > >   - Thank you for the suggestion. In the revision, we add a brief note in Sec. 2 that many perturbation-based explainers use entropy regularization to encourage near-binary masks. Sec. 3.4 then provides the details.
> > > - **Q10: Clarification about explanation results for dipole moment.**
> > >   - Thank you for this helpful comment. We agree that the dipole moment $\mu$ is a global vector observable and depends on both (i) local polarity/charge separation and (ii) the geometric separation of these charges. Our intention in Fig. 4 is therefore not to claim that a small subset of atoms “fully determines” the dipole moment, but rather to visualize which chemically meaningful functional groups are identified as most influential under a fixed node budget, using the same protocol across explainers.
> > >   - Regarding the suggestion to predict partial charges:
> > >     - To the best of our knowledge, most widely used 3D molecular GNN backbones [3-6], including SchNet and DimeNet++, predict global molecular properties (including dipole moment) in the QM9 dataset directly from atom types and 3D geometry, without using partial-charge labels as inputs or predicting charges as intermediate outputs.
> > >     - As a result, we do not have a standard backbone that would enable a widely recognized charge-based comparison. Therefore, we focus on explaining these common backbone designs and addressing the fidelity issues faced by current explainers, and leave incorporating additional domain knowledge (including partial charges) to future work.
> > >
> > > References
> > >
> > > [1] Molecular Modelling: Principles and Applications, Leach et al., Pearson, 2001.
> > >
> > > [2] Generalized neural-network representation of high-dimensional potential-energy surfaces, Behler et al., PRL 2007.
> > >
> > > [3] Provably powerful graph networks, Maron et al., NeurIPS 2019.
> > >
> > > [4] Spherical message passing for 3D graph networks, Liu et al., arXiv 2021.
> > >
> > > [5] E(n) equivariant graph neural networks, Satorras et al., ICML 2021.
> > >
> > > [6] ComENet: Towards complete and efficient message passing for 3D molecular graphs, Wang et al., NeurIPS 2022.

---

> > > > ### Comment · Reviewer_jjhE · 2026-03-24
> > > >
> > > > Dear authors,
> > > >
> > > > thank you for your detailed rebuttal. Thank you for addressing my comments regarding baseline results, how fidelity changes with K, why predicting 2 energy values is important, how T is controlled and why EDMA can be seen as a form of entropy regularization already and it would not add additional benefits to combine the two.
> > > >
> > > > Regarding the transfer to larger molecules, it can be assumed, that the number of edges in cutoff graphs scales linearly (not quadratically) with the number of atoms. Your example using a large cutoff of 10 Angstrom for ~1000 atoms illustrates this nicely: assuming the average neighborhood size is 20 atoms for this large cutoff, then we get 20*1000=20,000 edges in total.
> > > >
> > > > Considering the very large gap between Fidelity- and baseline MAE (for example Fidelity- > 28 eV difference in Table 4 vs. MAE of 0.007 eV would be a > 4000x increase in error), I am still a bit unsure if the predicted targets you propose can be explained very well using a subgraph explanation strategy at all: Free energy is an extensive property and heavily depends on the number of atoms, which is an information not available to your model since the number of nodes in the subgraph is a hyperparameter. Similarly the dipole moment depends on the "scaffold" of all atoms like discussed before. I still acknowledge that your proposed method outperforms the considered explanation baselines.

---

> > > > > ### Author Response · Authors · 2026-03-28
> > > > > **Second Round Responses to Reviewer jjhE**
> > > > >
> > > > > Thank you for your response and for your willingness to engage in discussion. We greatly appreciate your feedback and would like to offer a few clarifications.
> > > > >
> > > > > - **Q1: Relationship between the number of cutoff edges and atoms.**
> > > > >
> > > > >     Thank you for the response. Under a fixed cutoff radius, the edge number is governed by the average neighborhood size $k$: $|E|\approx O(nk)$, where $n$ is the number of atoms and $k$ is the average number of neighbors per atom. In large-graph settings, $k$ is typically bounded (locally dense), whereas for small molecules common cutoffs can make $k$ comparable to $n$ (nearly dense).
> > > > >     - This is the regime we observe on EC: graphs are locally dense but large. Even with $k$ on the order of tens, a graph with $\sim 10^3$ atoms can still contain $\sim 10^4–10^5$ edges, which is sufficient to trigger the density-amplification effect in our analysis. In particular, our bound depends on $d_{\max}$, which can be large in locally dense graphs.
> > > > >     - For small molecules such as QM9, $n$ is small and common cutoff choices (e.g., $5 \text{Å}$ or $10 \text{Å}$) can make many atoms mutual neighbors, so the cutoff graph can become nearly complete and $|E|$ approaches $O(n^2)$.
> > > > >     - Our analysis targets fidelity loss induced by dense graphs, so it applies to both “small but nearly dense” (QM9) and “large and locally dense” (EC). **We use QM9 as a widely adopted 3D GNN benchmark to illustrate the core challenges**, and include EC to verify that the effect persists in large-graph regimes.
> > > > >
> > > > > - **Q2: Applicability of subgraph explanations to global and extensive molecular properties.**
> > > > >     - **While properties such as free energy and dipole moment depends heavily on the full atomic structure, it does not mean that every part contributes equally, nor that subgraph explanations are chemically meaningless.** For example, many molecules contain dominant polar functional groups that strongly influence the dipole moment, and local environments can contribute disproportionately to learned energy-related targets. **This is exactly what graph explanation means, we aim to find the substructures that strongly influence the model prediction.**
> > > > >
> > > > >     - To further support our claim, we compare explanatory subgraphs produced by our method with randomly selected subgraphs under the same node budgets. If no meaningful substructure existed, EDMA would perform similarly to random selection. Instead, EDMA consistently outperforms random sampling for both $\mu$ and $G_f$ (Tables 1–2 below), suggesting that the backbone models place higher importances on specific substructures.
> > > > >
> > > > >     Table 1: Explanation $\mbox{Fidelity}^{-}$ (the lower the better) on the property $\mu$ (dipole moment, in Debye (D)) with SchNet as the fixed 3D GNN backbone being explained, EDMA vs. random subgraphs.
> > > > >
> > > > >     | $K$| 2 |  3 |4 |  5 |6 |  7|8 |  9 |
> > > > >     | --- | --- | --- | --- | --- | --- | --- | --- | --- |
> > > > >     | EDMA | 2.74 | 3.73 | 4.31| 4.83 |5.08 | 5.47 | 5.72 | 5.31 |
> > > > >     |   Random | 5.28 |8.71  | 11.15 | 12.16 |12.63  |13.44 | 13.00 |13.08 |
> > > > >
> > > > >     Table 2: Explanation $\mbox{Fidelity}^{-}$ (the lower the better) on the property $G_f$ (free energy, in eV) with SchNet as the fixed 3D GNN backbone being explained, EDMA vs. random subgraphs.
> > > > >     | $K$| 2 |  3 |4 |  5 |6 |  7|8 |  9 |
> > > > >     | --- | --- | --- | --- | --- | --- | --- | --- | --- |
> > > > >     | EDMA | 8.66| 7.45 | 6.23| 5.07 |3.74 | 2.55 | 1.36 | 0.21 |
> > > > >     | Random | 11.12 |10.63  | 10.07 | 9.51 |8.98  |8.37 | 7.80 |7.04 |

---

### Review · Reviewer_sUKg · 2026-02-27

**Summary Of Contributions:**

The paper introduces a new explainability approach designed for 3D graph neural networks. The Authors note that most existing explainable AI (XAI) methods are tailored for 2D structures and tend to underperform on 3D data such as molecular conformations, mainly because of dense graph connections. To address these challenges, they introduce an energy-based mechanism that creates discrete masks by learning two energy values per node, which are then converted into explanation masks. Their optimization loss includes a standard term ensuring that the masked graph still yields accurate predictions, and a second term that minimizes the difference between the energy values. The approach is evaluated on two molecular datasets involving small molecules and proteins, using fidelity metrics. The results are supported by qualitative analyses and an ablation study.

Strengths:
1. The method tackles the challenge of explaining 3D GNN predictions, an underexplored but highly important topic given the recent surge of interest in equivariant neural networks.
2. To my knowledge, applying energy-based modeling to explain GNNs is a novel approach, representing a different paradigm compared to methods based on gradients or information theory.
3. Section 4 demonstrates that EDMA significantly surpasses the other three methods on the selected benchmarks.

Weaknesses:
1. Some parts of the text lack clarity. For example, the sentence "We identify key sources of explanation errors and decompose them into two components, derived from an upper bound between optimized masks and the true explanatory subgraph" can be confusing because it includes many terms that were not explained, such as "two components" or "optimized masks".
2. The reference to Erdős and Rényi on page 5 is broken, and the bibliography entry should also be corrected.
3. The two issues related to GNN explanation methods discussed on page 5 are not addressed when describing the EDMA contribution. I believe not all 2D graph explanation methods assume edge randomness or independence - some may do so indirectly, but there are approaches that do not rely on this assumption. How does EDMA differ in this context? I do not understand how the assumption is removed using the loss function in Equation 7, compared to other methods with a similar loss function shown in Equation 8.
4. Similarly, I do not see what exactly makes the proposed method more robust for dense graphs. The issue 2 vaguely describes that dense graphs introduce "a substantial lack of confidence," which seems to be unsupported by any evidence. This observation should be supported by experiments that test different levels of graph density and compare the other methods to EDMA, or a reference should be provided for this claim.
5. Claiming that 3D GNNs have "exponentially many edges" is inaccurate. The number of edges grows quadratically as the number of nodes increases, so this statement is an exaggeration. Perhaps the intended focus was on the number of subgraphs?
6. The Authors claim that other methods that use the optimization objective presented in Equation 8 need to carefully tune their hyperparameters, but EDMA actually includes more hyperparameters (T, $\zeta$, $\gamma$, $\alpha$). The Authors do not discuss how these parameters are tuned. What values should be tested? What are the final hyperparameters used?
7. The quantitative results are limited to the fidelity metric evaluated on only two datasets. The comparison includes just three baselines, and several key explainability methods, such as gradient-based techniques, are absent. Why does Table 8 include only one other baseline? Additionally, the code is not available, which hampers the reproducibility of the results.
8. I do not understand the "Chemical Explanation" column in Figure 4. It appears that those atoms and bonds were arbitrarily chosen to fit the explanation, and other electronegative atoms could be selected to fit a different hypothesis. What was the exact procedure in producing the "Chemical Explanation" column?

In summary, the presented method addresses an important problem of explaining 3D graph models. The idea seems novel, and the experiments demonstrate strong results on the fidelity metric, but some claims in the paper need more supporting evidence.

**Audience:**

Yes

**Audience Explanation:**

3D GNNs and equivariant neural networks are now the primary architectures employed for analyzing molecular data and are also applicable in other domains. Developing effective explanation methods for their predictions is essential to progress ML research in scientific discovery.

**Broader Impact Concerns:**

I think this work does not require an extensive description of broader impacts. However, the Authors could consider discussing the limitations of their method and how graph explainers might create false trust in the results, which is especially important for AI in science.

**Claims And Evidence:**

No

**Claims Explanation:**

As described in the summary:
1. The problems with XAI for GNNs mentioned on page 5 require additional theoretical or empirical evidence to confirm both the existence of these problems and that EDMA addresses them effectively.
2. The claims about having fewer hyperparameters on page 7 and the recurring phrase "exponentially many edges" need correction.
3. The Chemical Explanation column in Figure 4 appears to be selected arbitrarily, so it does not offer a reliable ground truth.
4. The evaluation is limited to two datasets and three baseline methods, and the code is not provided, which reduces confidence in the robustness of the results.

**Requested Changes:**

Please refer to the weaknesses in the summary.

Most importantly, all claims in the paper should be backed by references or sufficient experiments. The method's motivation needs more support, such as a clear statement of how EDMA solves the two problems on page 5. Since code is not provided, the experimental section should evaluate EDMA more thoroughly, and all hyperparameters should be reported. A possible extension could include the remaining QM9 tasks and additional types of explainers, like Integrated Gradients (gradient-based).

---

> ### Author Response · Authors · 2026-03-20
> **Response to Reviewer sUKg - Part I**
>
> Thank you so much for your comments! We have revised the manuscript and provide our responses below.
>
> - **W1: Clarity of terms, such as "two components" and "optimized masks".**
>
>   - We thank the reviewer for noting this ambiguity and would like to clarify. **We have revised the Abstract accordingly to make these terms explicit and avoid ambiguity.**
>     - **The term “optimized masks” refers to the soft masks learned by gradient-based optimization in most explainers.** In graph explanation, the goal is to output a small discrete subgraph (a subset of nodes/edges) whose prediction faithfully represents the full graph's behavior. Since directly optimizing discrete subsets is not differentiable, existing methods typically introduce a continuous relaxation: they assign soft masks to nodes/edges, optimize these soft masks via gradient descent, and then discretize the optimized soft masks to obtain the final explanatory subgraph.
>     - **The “two components” refer to the two terms in the upper bound in Eq. (5) that arise from this relaxation-and-discretization pipeline**: (i) the optimized soft-mask loss (the loss achieved by the continuous relaxation), and (ii) the soft-to-discrete discrepancy introduced when discretizing the optimized soft mask to form the explanatory subgraph. Our theoretical analysis shows that the second term can be amplified in dense 3D graphs, which motivates EDMA: we use an energy-based parameterization designed to reduce both components simultaneously.
>
> - **W2: The broken reference and bibliography.**
>
>   - Thank you for pointing this out. We have fixed the broken citation link and regenerated the bibliography to ensure all references resolve correctly in the updated manuscript.
>
>
> - **W3: Two issues related to GNN explanation.**
>   - **Two issues in explaining 3D graphs.**
>     - In 3D molecular GNNs, edges are typically induced from geometry (e.g., cutoff neighbors) rather than being fixed chemical bonds, which often yields much denser graphs. This creates two challenges when applying common perturbation-based 2D explanation methods to 3D GNNs.
>     - First, many widely used perturbation-based explainers (e.g., GNNExplainer/PGExplainer) optimize one mask variable per edge, effectively treating edge decisions as independent during optimization. In 3D cutoff graphs, however, edges are strongly coupled because adding or removing an atom simultaneously affects many incident edges.
>     - Second, increased edge density amplifies the soft-to-discrete discrepancy when converting optimized soft masks into a discrete explanatory subgraph, which can reduce explanation fidelity. Eq. (5) and our analysis in Appendix A explain why dense 3D graphs are particularly challenging when high-fidelity explanations are desired.
>     - **We have revised Sec. 3.1 to state these issues more precisely.**
>
>
>   - **How does EDMA differ in this context.**
>     - EDMA addresses these issues through its mask parameterization and discreteness mechanism:
>       - Instead of optimizing independent edge masks, EDMA optimizes node masks and induces edge masks via node-to-edge coupling, aligning explanations with how edges are generated in 3D graphs.
>       - EDMA’s energy-gap and temperature formulation is designed to drive node masks toward near-binary values, reducing the soft-to-discrete discrepancy that becomes more severe in dense cutoff graphs (formalized in Appendix B and reflected in the experiments).
>
>   - **Does EDMA remove the assumption via Eq. (7) vs Eq. (8)?**
>     - EDMA does not rely on “just” changing the loss from Eq. (8) to Eq. (7). The key difference is what is optimized: Eq. (8) optimizes soft masks (often one variable per edge in the original formulations), whereas EDMA optimizes node-level masks and induces edge masks through node coupling using an energy-based formulation. Eq. (7) is the objective used to optimize this node-gated parameterization to achieve high-fidelity explanations on dense 3D graphs.
>     - Together, node-induced masking (addressing edge coupling) and the energy-gap/temperature mechanism (addressing the soft-to-discrete discrepancy under density amplification) explain why EDMA is more robust on dense 3D graphs.

---

> > ### Author Response · Authors · 2026-03-20
> > **Response to Reviewer sUKg - Part II**
> >
> > - **W4: Evidences to support the claim of "a substantial lack of confidence"**
> >   - To address this concern, **we added Appendix A and Appendix B** to make the “lack of confidence” statement precise and to explain why EDMA is more robust on dense graphs.
> >   - **What we mean by “lack of confidence”.**
> >     - By “lack of confidence,” we mean that optimization-based explainers often converge to fractional soft masks (many values in (0, 1)) rather than near-binary masks. Since the final explanation is obtained by discretizing these optimized soft masks into a Top-$K$ discrete subgraph, the resulting discrete masks can deviate substantially from the soft masks.
> >     - This issue becomes more pronounced in dense 3D graphs because each atom participates in many cutoff edges, so fractional node masks affect a large portion of message passing.
> >
> >   - **Why density amplifies the soft-to-discrete discrepancy.**
> >     - Appendix A provides a quantitative bound that links graph density to the soft-to-discrete prediction gap. Let $\hat Y(\mathbf M')$ denote the backbone’s prediction when message passing is masked by a soft mask $\mathbf M'\in[0,1]^{n\times n}$, and let $\hat Y(\mathbf M)$ denote the prediction under the corresponding discrete mask $\mathbf M\in\lbrace0,1\rbrace^{n\times n}$ after Top-$K$ discretization. In our setting, $\mathbf M'=\boldsymbol m\otimes \boldsymbol m$ with node masks $\boldsymbol m\in[0,1]^n$ and its discretization $\mathbf M=\boldsymbol m^d\otimes \boldsymbol m^d$ with $\boldsymbol m^d\in\lbrace0,1\rbrace^n$. Appendix A shows the discrepancy term in Eq. (5) admits the following explicit upper bound (with $L_\Phi$ a finite Lipschitz constant and $d_{\max}$ the maximum node degree):
> >   $
> >   \big|\hat Y(\mathbf M')-\hat Y(\mathbf M)\big|
> >   \le
> >   2L_\Phi d_{\max}\|\boldsymbol m-\boldsymbol m^d\|_1.
> >   $
> >
> >      - The factor $d_{\max}$ formalizes the density amplification effect: when the graph is dense (large $d_{\max}$), even modest non-binary mask values can be strongly amplified, making it harder for soft-mask optimization to translate into faithful discrete subgraphs.
> >
> >
> >   - **Why EDMA is more robust in this regime.**
> >     - Appendix B shows EDMA’s energy-gap parameterization directly controls the soft-to-discrete distance. Let $\Delta\mathcal{E}_i=\mathcal{E}_1(\mathbf e_i)-\mathcal{E}_0(\mathbf e_i)$ and temperature $T>0$. Appendix B gives a bound of the form
> >
> >        $
> >       \big|\hat Y(\mathbf M')-\hat Y(\mathbf M)\big|
> >       \le
> >       2L_\Phi d_{\max} \sum_{i=1}^n\exp\bigl(-\Delta\mathcal{E}_i/T\big).
> >       $
> >
> >     - Thus, annealing $T$ (or increasing energy gaps) pushes masks rapidly toward {0, 1}. Combined with Appendix A, this shows EDMA directly tightens the discrepancy term in Eq. (5). Therefore, EDMA is more robust on dense 3D graphs than baselines.
> >
> > - **W5: The claim that 3D GNNs have "exponentially many edges".**
> >   - Thank you for catching this. The number of edges grows quadratically with the number of nodes. **We have revised the manuscript accordingly.**
> > - **W6: Hyperparamete settings.**
> >   - We thank the reviewer for pointing this out. EDMA introduces four hyperparameters: $\gamma, \zeta, \alpha$, and the temperature $T$. We now describe how they are set and tuned, and **we have modified the manuscript and list the final values in Appendix D.**
> >     - **Stretching parameters $\gamma,\zeta$**. We introduce $\gamma<0$ and $\zeta>1$ as optional parameters to control the shape of the relaxed masks: mapping probabilities to $(\gamma,\zeta)$ and clamping to $[0,1]$ can encourage faster saturation toward {0, 1} and provides a simple mechanism to adjust discreteness across settings if needed. Empirically, EDMA performs robustly with a single default choice, so we do not tune $\gamma,\zeta$ and fix them to $\gamma=-0.1, \zeta=1.1$ in all experiments.
> >     - **Budget coefficient $\alpha$.** $\alpha$ balances the prediction loss and the budget term. We tune $\alpha$ via a small grid search on the validation split (same protocol as baselines), selecting the value that minimizes validation explanation error under the fixed budget $K$. We report test results using the selected $\alpha$. The final $\alpha$ values for each dataset and backbone are reported in Appendix D.
> >     - **Temperature schedule $T$.** $T$ controls mask discreteness during optimization: higher $T$ produces smoother masks early on (stabilizing gradients), while annealing $T$ sharpens the sigmoid gating and drives masks toward {0, 1}, reducing the soft-to-discrete discrepancy. In all experiments, we fix the initial temperature $T_0=5.0$ and use geometric decay with a mild exponent ($0.4$). We additionally apply an epoch cap to stop annealing once masks are already near-binary to avoid over-annealing. We choose the cap by a small validation-based search, and use: $\text{cap} =4$ for $\mu$ with SchNet, $\text{cap} =25$ for $G_f$ with SchNet, $\text{cap} =25$ for $\mu$ with DimeNet++, and $\text{cap} =36$ for $G_f$ with DimeNet++.

---

> > > ### Author Response · Authors · 2026-03-20
> > > **Response to Reviewer sUKg - Part III**
> > >
> > > - **W7: Limited datasets, more baseline comparison and the code availability.**
> > >   - **Scope of quantitative evaluation (two datasets, fidelity metric).**
> > >     - Our paper studies explanation fidelity for 3D molecular GNNs under cutoff-based graphs.
> > > **We evaluate on QM9 (small molecules where cutoff graphs can be dense) and EC (substantially larger graphs with locally dense neighborhoods), which together cover both the “small but dense” and “large and locally dense” settings relevant to the density amplification effect analyzed in the paper.**
> > >     - On QM9, there are multiple properties, and each property corresponds to a different prediction problem, and standard pretrained pipelines train a separate model per target. We therefore report results on two important and representative targets in chemistry, dipole moment $\mu$ and free energy $G_f$, using pretrained SchNet and DimeNet++ backbones, to demonstrate the phenomenon across both properties and architectures under the same evaluation protocol.
> > >     - We also note that these 3D backbones are commonly evaluated on a small set of molecular benchmarks (e.g., QM9 for regression tasks and one or two molecular dynamics datasets). We evaluate on QM9 for small molecules and additionally include the EC dataset to assess generalization to larger graphs. For evaluation metric, we use $\mbox{Fidelity}^{-}$ for regression (QM9) and $\mbox{Fidelity}^{+}$ for classification (EC), and we report results across multiple budgets $K$ for a comprehensive evaluation.
> > >   - **Baseline comparisons.**
> > >     - Our focus is the **soft-to-discrete fidelity issue in optimization-based explainers on dense 3D graphs**. We therefore compare against representative and widely used mask-optimization baselines (GNNExplainer and PGExplainer) and include LRI as a 3D explainer.
> > >     - Gradient-based saliency methods typically produce continuous attributions rather than a budgeted discrete subgraph; converting them into Top-$K$ subgraphs requires additional thresholding, which would introduces additional confounds beyond the relaxation-and-discretization effect studied here.
> > >     - We further add GEM[1] as a discrete selection baseline to isolate the relaxation effect. Together, these baselines cover the most relevant models for the issue analyzed in our paper while remaining comparable under the same Top-$K$ evaluation protocol.
> > >       - GEM is a causal explanation approach (motivated by Granger causality) that incrementally selects edges to form an explanatory subgraph. Importantly, GEM performs discrete selection rather than optimizing soft masks, and thus avoids the relaxation-induced soft-to-discrete discrepancy. Since GEM is originally formulated for the edge selection, we adapt it to the node selection to ensure a fair comparison.
> > >       - **Table 1 shows that EDMA outperforms GEM for most budgets $K$ on SchNet.** It further demonstrates the faithful explanation predicted by EDMA.
> > >     Table 1: Explanation $\mbox{Fidelity}^{-}$ (the lower the better) on the property $\mu$ (dipole moment, in Debye (D)) with SchNet as the fixed 3D GNN backbone being explained, comparing baselines, GEM, and EDMA. For each $K$, the best results are boldfaced, and the second-best results are marked with $^{\dagger}$.
> > >    |K|2|3|4|5|6|7|8|9|
> > >    |---:|---:|---:|---:|---:|---:|---:|---:|---:|
> > >    |GNNExplainer-Dense| 3.88 | 5.62 | 7.28 | 8.05 | 8.27 | 8.00 | 7.59 | 6.87 |
> > >    |PGExplainer-Dense | 2.91 | 3.73$^{\dagger}$ | 4.83 | 6.09 | 6.62 | 6.55 | 6.81 | 6.08|
> > >    |LRI-Bernoulli | 3.50 | 4.84 | 6.16 | 6.88 | 7.10 | 7.29 | 7.43 | 7.32 |
> > >    | GEM| 2.76$^{\dagger}$| **3.66** | 4.55$^{\dagger}$ | 5.13$^{\dagger}$ | 5.69$^{\dagger}$ | 6.26$^{\dagger}$ |6.31$^{\dagger}$ | 5.89$^{\dagger}$ |
> > >    |**EDMA** | **2.74** | 3.73$^{\dagger}$ |**4.31**| **4.83**| **5.08** | **5.47** |**5.72**| **5.31**|
> > >
> > >   - **Results in Table 8.**
> > >     - Table 8 is an additional experiment to check whether EDMA also works with an equivariant backbone (EGNN), not only invariant backbones (SchNet and DimeNet++). The main purpose of this experiment is to demonstrate the genralizability of EDMA. Therefore, we only modify the GNNExplainer to be applicable on equivariant model, which is then used as a representative optimization-based baseline. Therefore, Table 8 proves that our EDMA could be applied on equivariant models with competitive performance.
> > >
> > >   - **Code availability.**
> > >     - We have released an anonymized code repository for the submission:  https://anonymous.4open.science/r/EDMA-9EDA/.

---

> > > > ### Author Response · Authors · 2026-03-20
> > > > **Response to Reviewer sUKg - Part IV**
> > > >
> > > > - **W8: "Chemical Explanation" in Figure 4.**
> > > >
> > > >   - Thank you for raising this concern. We construct the “Chemical Explanation” column in Fig. 4 using a two-step procedure.
> > > >   - First, we rely on domain experts in chemistry to select the key functional groups relevant to the dipole moment from candidate functional groups extracted from the provided 3D molecular structures. For each molecule, experts provide a brief chemistry-based rationale to justify selecting these functional groups, and we present these annotations in the “Chemical Explanation” and “Functional Group” columns. These annotations are produced independently of the model explanations (experts did not see EDMA/baseline outputs) and follow standard chemical heuristics for the dipole moment.
> > > >     - Regarding the concern that the “Chemical Explanation” may be arbitrary: while multiple electronegative atoms may exist in a molecule, the dipole moment is typically dominated by the strongest polar bonds or functional groups and their local geometric environment. The expert selections therefore prioritize the dominant polar functional groups that create the largest charge separation and contribute most to the molecular dipole, rather than selecting atoms post hoc to match any particular explainer output.
> > > >    - Second, we run each baseline and EDMA to produce an explanatory subgraph. To ensure a fair comparison with the expert-provided functional group, we set the node budget for each method to match the number of atoms in the corresponding candidates. We then visualize the resulting substructure by connecting the selected atoms using the original chemical bonds in QM9.
> > > >    - Overall, this protocol allows us to assess whether EDMA more consistently recovers the expert-identified dominant polar functional groups under the same node budget, compared to baselines.
> > > >
> > > > References
> > > >
> > > > [1] Generative causal explanations for graph neural networks, Lin et al., ICML, 2021.

---

> > > > > ### Comment · Reviewer_sUKg · 2026-03-29
> > > > >
> > > > > Thank you for your response! I have read all the replies, the updated manuscript, and the appendix. I want to clarify two final issues before I make my recommendation:
> > > > >
> > > > > 1. The lemmas and theorems employ numerous inequalities, resulting in an estimate of only the upper bound, likely with some margin of safety. While we understand that the upper bound scales with the maximum node degree, this does not mean that the prediction difference is directly proportional to it. Also, would you agree that, given (A.25), the $L_\Phi$ term is driving the upper bound in deep architectures more than the node degree is?
> > > > > 2. While the reasoning behind the selection of important atoms by chemistry experts, as explained in the response, seems reasonable, I still believe that the presented results are arbitrary for two reasons. First, no details are provided about the number of experts and the number of labeled compounds. Is it only for the three compounds shown in Figure 4? If more compounds were labeled, then the selection of compounds could also be arbitrary, and all labeled compounds should be included in the appendix. Also, could you confirm whether the domain expert in the third example in Figure 4 selected the -NH group with only one hydrogen atom (in line with the model prediction), even though the structure has the -NH2 group?

---

> > > > > > ### Author Response · Authors · 2026-04-01
> > > > > > **Second Round Responses to Reviewer sUKg**
> > > > > >
> > > > > > Thank you so much for your response and for your willingness to engage in discussion. We greatly appreciate your feedback and would like to offer a few clarifications.
> > > > > >
> > > > > > - **Q1: On looseness of the bound in Appendix A (Eq. A.25)**
> > > > > >     - **Worst-case nature of the bound.**
> > > > > >     We agree that our derivation provides a worst-case upper bound, and it should not be interpreted as implying that the prediction gap is directly proportional to $d_{\max}$. Our purpose is not to estimate a tight constant, but to **isolate structural factors that can systematically enlarge the soft-to-discrete discrepancy in cutoff graphs.**
> > > > > >     - **Why the degree factor is still meaningful.**
> > > > > >     The dependence on $d_{\max}$ in Eq. A.25 arises from uniformly controlling neighborhood summations (bounding $\sum_{j\in\mathcal N(i)}$ by $d_{\max}$ times a node-level perturbation scale). This captures a genuine accumulation mechanism: in denser cutoff graphs, the same node-level soft-to-discrete discrepancy affects more incident cutoff edges, so the induced edge-mask perturbation can grow with neighborhood size. This conclusion is structural and does not rely on the bound being tight.
> > > > > >     - **On depth-dependent terms in Eq. A.25.**
> > > > > >     We agree that for deep architectures, the unrolled product terms in Eq. A.25 can dominate the numerical value of the bound, making it conservative. However, this does not contradict our density argument.
> > > > > >         - Eq. A.25 shows that $L_\Phi$ can grow with both network depth and neighborhood aggregation, and **density enters $L_\Phi$ through the neighborhood aggregation mechanism (see Eq. A. 23)**, providing an additional amplification channel for the soft-to-discrete discrepancy even when depth also contributes to conservativeness.
> > > > > >
> > > > > >
> > > > > >
> > > > > > - **Q2: The chemical explanation in Fig. 4**
> > > > > >
> > > > > >     - Consistent with the standard perturbation-based setting and the baselines, our goal is not to claim that a small subgraph fully determines a molecular property, but to **identify a compact Top-$k$ substructure whose prediction best approximates the full-graph prediction.**
> > > > > >     - Fig. 4 is intended as a qualitative illustration of dominant polar functional groups for dipole moment.
> > > > > >         - Since QM9 does not provide ground-truth explanatory subgraphs, we consulted chemists to annotate a set of molecules; given the scale of QM9 ($≈134k$ molecules), it is not feasible to annotate and include all of them.
> > > > > >     - Our core contribution is **analyzing and reducing the soft-to-discrete discrepancy in optimization-based explainers on dense 3D cutoff graphs.** EDMA’s reduced discrepancy makes it more likely to produce confident discrete substructures that align with chemical intuition.
> > > > > >     - However, Fig.4 could be misread as a mechanistic claim about how the backbone “truly” computes $\mu$, which is beyond our scope.
> > > > > >         - To avoid confusion, we are considering moving Fig. 4 to the appendix and adding a detailed description in the revision. We would appreciate the reviewer’s preference: (i) keep Fig. 4 in the main paper and refer readers to details in the appendix, or (ii) move Fig. 4 (together with the detailed description) to the appendix. We will revise accordingly based on your suggestion.
> > > > > >     - Regarding the third example, the expert annotation was at the functional-group level, **highlighting the amino site (the $N$ atom and its $N–H$ bond polarity) together with the nearby carbonyl group**, which often makes a dominant contribution to $\mu$. The intent was to highlight the dominant polar region rather than the exact number of hydrogens.

---

> > > > > > > ### Comment · Reviewer_sUKg · 2026-04-03
> > > > > > >
> > > > > > > Thank you for the clarification!
> > > > > > >
> > > > > > > I see value in providing examples of explanations and comparing them to the results from other explainers. I also believe that describing the functional groups important for calculating dipole moments is helpful for evaluating these results. I recommend keeping Figure 4 in the main paper, but with a clearer explanation of the "Chemical Explanation" column, a different way of representing these expert annotations, or moving just this column to the supplementary material. The text should at least include the total number of annotated examples, the number of experts and their backgrounds, and details of the labeling procedure (I suppose they did not see the model explanations before indicating the most important atoms and bonds).
> > > > > > >
> > > > > > > As for the representation of annotations, if the experts only highlighted important functional groups, would it be fair to highlight these groups in the full structure at the group level? Based on the description in the response, I do not believe it is fair to suggest that the annotator indicated specific atoms and bonds. Additionally, if you could expand the annotated set to at least 50 molecules (a suggestion for the camera-ready version, not for this discussion), this experiment might have enough statistical power to support the claims with quantitative results, such as the average overlap between explanations and annotations.
> > > > > > >
> > > > > > > Thank you again for your response!

---

> > > > > > > > ### Author Response · Authors · 2026-04-03
> > > > > > > > **Third Round Responses to Reviewer sUKg**
> > > > > > > >
> > > > > > > > - **Response to suggestions about Fig. 4**
> > > > > > > >
> > > > > > > > Thank you for the constructive suggestions. We will keep Fig. 4 in the main paper and revise it for clarity. Due to the limited time, we will implement the following changes in the camera-ready revision：
> > > > > > > > - Specifically, we will present the expert annotations at the functional-group level by highlighting the annotated groups on the full molecular structures. And we will also update the manuscript to clarify that the experts labeled functional groups rather than specific atoms/bonds.
> > > > > > > > - We will keep the visual comparison in the main paper, and move the detailed “Chemical Explanation” text (or that column) to the appendix. The labeling protocol (including clearly statement that experts did not see model explanations) and additional more annotated examples (under affordable resources) will be provided in the appendix.

---

### Decision · Action_Editor_eYKn · 2026-04-14

**Recommendation:** Accept with minor revision

**Additional Comments:**

This paper first discusses the dense graph connection problem when applying existing explanation methods to 3D molecules, and then proposed a novel energy-based method that remedies the problem.  Experiments on small molecular and protein datasets show consistent improvements in fidelity.

Reviewers generally acknowledged the novelty and significance, but raised some concerns, including clarity, weak theoretical support, missing baselines, and lack of analysis of efficiency and stability.  In addition, one reviewer initially questioned whether the claims were well supported.  The authors well addressed all concerns of the reviewers and revised the paper accordingly, except a few minor things that the authors promised to fix in the camera ready version.

Please make minor revision, as promised in discussion with Reviewer sUKg.

- Add the GEM result in the paper.
- Discussion on qualitative examples comparing explanation and chemical knowledge in Fig.4 and appendix.

**Audience:**

Yes

**Audience Explanation:**

All reviewers agree that the paper has audience in the ML community.

**Claims And Evidence:**

Yes

**Claims Explanation:**

A reviewer pointed out unsupported/incorrect claims, which have been addressed in the revision.  All reviewers agree that all claims in the revision are supported.